# divERGe implements various Exact Renormalization Group examples

Jonas B. Profe[1], Dante M. Kennes[1,2], Lennart Klebl[3*]

**1** Institute for Theory of Statistical Physics, RWTH Aachen University, and JARA
Fundamentals of Future Information Technology, Aachen, Germany
**2** Max Planck Institute for the Structure and Dynamics of Matter, Center for Free Electron
Laser Science, Hamburg, Germany
**3** I. Institute for Theoretical Physics, Universität Hamburg, Hamburg, Germany
*lennart.klebl@uni-hamburg.de

March 27, 2024

## Abstract

We present divERGe, an open source, high-performance C/C++/Python library for functional renormalization group (FRG) calculations on lattice fermions. The versatile model interface is tailored to real materials applications and seamlessly integrates with existing, standard tools from the *ab-initio* community. The code fully supports multi-site, multi-orbital, and non-$SU(2)$ models in all of the three included FRG variants: TU$^2$FRG, *N*-patch FRG, and grid FRG. With this, the divERGe library paves the way for widespread application of FRG as a tool in the study of competing orders in quantum materials.

# 1 Introduction

High-performance computing brought many insights into physical problems deemed unsolvable by analytic means. Especially the *ab-initio* treatment of condensed matter systems in the form of density functional theory, GW and dynamical mean-field approaches is a remarkable success story [1–7]. Besides the versatility of the methods mentioned above, a key ingredient for this success is the availability of optimized libraries, such as ABINIT [8], Quantum Espresso [9], Vasp [10], BerkleyGW [11], YAMBO [12], TRIQS [13] and many more [14], leveraging the need to develop complex codes over and over again. If no such public code is available, each researcher has to implement the method themselves, thus creating a lot of redundant work with, most likely, sub-optimal outcome. Therefore, it is elemental for the broad applicability of a method to have a public code available — as well as a documented knowledge about best practice in implementations.

In the study of correlated materials, the *ab-initio*-based treatments mentioned above already incorporates many important features. However superconducting orders from electronic interactions and other effects of long-range interactions generated during the approximate solution of the many-body Schrödinger equation are only partially, or not at all, included. This

creates the need for methods and codes that allow us to connect to these developments, and extend the status quo by adding the missing pieces. In computational condensed matter physics, it has proven efficient to start from an effective low energy description of the material, keeping only a few relevant bands. The process of how to arrive at such a down-folded model poses the first obstacle. Subsequently, we still have to solve a model including a few bands, with complicated interactions. To tackle such problems, we often need to introduce approximations, which should be well controlled. For a broad class of materials, we can utilize pertubative approaches, such as the random phase approximation (RPA), the parquet approximation [15] and FRG [16,17]. While the former one includes only specific diagrammatic channels, the latter two are diagrammatically unbiased, thus being prime candidates for the extension of the *ab-initio* machinery; the issue being that implementations of the full equations, incorporating all their dependencies are beyond our current reach.

In this paper we present divERGe[1] — an open source, high-performance (multi-node CPU & multi-GPU) C/C++/Python library (available at [18]) that implements different flavors of the FRG [16,17]. The library is based on a general model interface (cf. Section 3), and three different computation backends: (i) grid-FRG [19,20], (ii) truncated unity FRG (TU$^2$FRG) [21–23], and (iii) orbital space $N$-patch FRG [24–26]. Each performs different approximations of the central equations, resulting in different numerical complexity, as detailed in Appendix D. This paper is designed as a hands-on introduction to the usage of divERGe. We therefore briefly summarize FRG as a numerical method in Section 2, introduce the model structure in Section 3, explain how the flow equations are solved in Section 4 and how the results are analyzed in Section 5.

## 2   Theoretical Background

This paper is meant as an introduction into the usage of our library and is *not* meant to give a full introduction to functional renormalization group (FRG) as a computational method. For such an introduction we refer the reader to Refs. [16, 17, 27, 28]. Our library implements a level-2 truncated FRG in static vertex approximation and optional static self-energy feedback. This flavor of FRG, often called vertex flow (or RPA+), treats fluctuations from the different diagrammatic channels on equal footing. It thus allows for a diagamatically unbiased prediction of the phase diagram of a model. The method was widely applied to the 2D Hubbard model [21,22,25,29–31], and more recently to other, more complex, models [**?**,32–34,36–51]. For an overview of the three different flavors of the vertex flow FRG, we refer the reader to Ref. [52]. In the following we briefly explain the type of models that can be studied with divERGe, and thereafter detail the equations that are solved and give an introduction to the analysis of results.

### 2.1   General setup

In general, we aim to study arbitrary fermionic models, on arbitrary lattices, with a kinetic and a two-body interaction contribution. The Hamiltonian (in second quantization) for such a model reads

$$H = \underbrace{\sum_{12} t_{12} c_2^\dagger c_1}_{\hat{T}} + \underbrace{\sum_{1234} V_{1234} c_3^\dagger c_4^\dagger c_2 c_1}_{\hat{V}}, \tag{1}$$

with numbers denoting a collection of all quantum numbers specifying an electronic single-particle state. The model initialization from within the code is described in detail in Section 3.

---

[1]divERGe implements various Exact Renormalization Group examples

## 2.2 Flow equations

Once the model is implemented, we solve the flow equations for the three different diagramatic channels and (optionally) the static self-energy (primed variables are summed over):

$$\frac{d\Sigma_{1,3}^{\Lambda}(\boldsymbol{k})}{d\Lambda} = \sum_{2,4,\boldsymbol{k}'} S_{4,2}^{\Lambda}(\boldsymbol{k}') F_{1,2,3,4}^{\Lambda}(\boldsymbol{k}, \boldsymbol{k}', \boldsymbol{k}), \tag{2}$$

$$\frac{d\Phi_{1,2,3,4}^{pp,\Lambda}(\boldsymbol{k}_1, \boldsymbol{k}_2; \boldsymbol{k}_3)}{d\Lambda} = \frac{1}{2} F_{1,2,1',2'}^{\Lambda}(\boldsymbol{k}_1, \boldsymbol{k}_2; \boldsymbol{k}') F_{3',4',3,4}^{\Lambda}(\boldsymbol{k}', \boldsymbol{q}_P - \boldsymbol{k}'; \boldsymbol{k}_3) \dot{L}_{1',2',3',4'}^{\Lambda}(\boldsymbol{k}', \boldsymbol{q}_P - \boldsymbol{k}'), \tag{3}$$

$$\frac{d\Phi_{1,2,3,4}^{ph,\Lambda}(\boldsymbol{k}_1, \boldsymbol{k}_2; \boldsymbol{k}_3)}{d\Lambda} = F_{1,4',3,1'}^{\Lambda}(\boldsymbol{k}_1, \boldsymbol{k}'; \boldsymbol{k}_3) F_{3',2,2',4}^{\Lambda}(\boldsymbol{k}' - \boldsymbol{q}_D, \boldsymbol{k}_2; \boldsymbol{k}') \dot{L}_{1',2',3',4'}^{\Lambda}(\boldsymbol{k}', \boldsymbol{k}' - \boldsymbol{q}_D), \tag{4}$$

$$\frac{d\Phi_{1,2,3,4}^{cph,\Lambda}(\boldsymbol{k}_1, \boldsymbol{k}_2; \boldsymbol{k}_3)}{d\Lambda} = -F_{1,4',1'4}^{\Lambda}(\boldsymbol{k}_1, \boldsymbol{k}' - \boldsymbol{q}_C; \boldsymbol{k}') F_{3',2,3,2'}^{\Lambda}(\boldsymbol{k}', \boldsymbol{k}_2; \boldsymbol{k}_3) \dot{L}_{1',2',3',4'}^{\Lambda}(\boldsymbol{k}', \boldsymbol{k}' - \boldsymbol{q}_C), \tag{5}$$

where Matsubara frequency summations over $ik_0'$ are implicit. The full two particle vertex $F$ is given as

$$F^{\Lambda} = U + \Phi^{pp,\Lambda} + \Phi^{ph,\Lambda} + \Phi^{cph,\Lambda}, \tag{6}$$

with $U$ the fully irreducible vertex. The scale derivative of the loop is given in terms of the single particle Greens-function and the single scale propagator $S = G(\partial^{\Lambda}(G_0)^{-1})G$:

$$\dot{L}_{1,2,3,4}^{\Lambda}(\boldsymbol{k}_1, \boldsymbol{k}_2, \boldsymbol{k}_3, \boldsymbol{k}_4) = \left[ S_{1,3}^{\Lambda}(\boldsymbol{k}_1) G_{2,4}^{\Lambda}(\boldsymbol{k}_2) + G_{1,3}^{\Lambda}(\boldsymbol{k}_1) S_{2,4}^{\Lambda}(\boldsymbol{k}_2) \right] \delta_{\boldsymbol{k}_1, \boldsymbol{k}_3} \delta_{\boldsymbol{k}_2, \boldsymbol{k}_4}. \tag{7}$$

The non-interacting Green's function $G_0$ is modified by a multiplicative regulator $f(\Lambda)$:

$$G_0 \rightarrow G_0^{\Lambda} = G_0 f(\Lambda), \tag{8}$$

and the interacting Green's function is given as

$$G^{\Lambda} = \left( (G_0^{\Lambda})^{-1} - \Sigma^{\Lambda} \right)^{-1}. \tag{9}$$

Within divERGe, we employ a sharp cutoff as regulator, i.e., $f(\Lambda) = \Theta(|\omega| - \Lambda)$. This choice significantly reduces numerical complexity in several parts of the code (see Ref. [52] and Appendix D).

From the above equations, we immediately find the connections between FRG and other diagrammatic approaches, such as the RPA and the parquet approximation. If we were to neglect the flow of all but one channel, we restore the RPA series specific to that channel [28], while when comparing with parquet, we find that we miss the multi-loop class of diagrams [53–55]. These observations motivate a nomenclature as "RPA+" of the vertex flow — we include the RPA diagrams of all channels and most of the simultaneous cross-insertions of those ladders as feedback (missing only the multi-loop corrections).

## 2.3 Analysis of results

The differential equations for the vertices (flow equations) are solved numerically until a divergence of one of the vertex components occurs or the minimal scale is reached. A divergence is indicative of a phase transition to an ordered state. In that case, the flow is stopped and from analysis of the vertex, susceptibilities, and linearized gap equations, we can extract information of the ordered state, see Ref. [52].

## 3 Creating a Model

The central object in a momentum space FRG calculation is a model that includes information on kinetics and interactions. Alongside with some more parameters, these two pieces of information make the required ones to setting up a simulation. For defining a model from a Hamiltonian, we first have to choose our basis states

$$|\mathbf{R}, o_1, s_1\rangle, \tag{10}$$

with $o_1$ the combined orbital-site index within a unit cell and $\mathbf{R}$ the real-space lattice vector which connects a site to the reference unit cell. The spin index in $z$ basis is denoted by $s_1$. While the code technically allows for arbitrary spins, the spin symmetric variant of the flow equations is only implemented for spin-1/2 particles.

### 3.1 Required Structures

We choose to use a formulation of kinetics based on real space hopping parameters. Qualitatively equivalent, the vertices are also passed as real space description in the three distinct interaction channels. Both must be supplied by the user. The following section explains these three key ingredients of the code, starting with the overarching `diverge_model_t` structure:

```
1 struct diverge_model_t {
2     char name[MAX_NAME_LENGTH];
3
4     index_t nk[3];
5     index_t nkf[3];
6     mom_patching_t* patching;
7
8     index_t n_ibz_path;
9     double ibz_path[MAX_N_ORBS][3];
10
11     index_t n_orb;
12     double lattice[3][3];
13     double positions[MAX_N_ORBS][3];
14
15     index_t n_sym;
16     complex128_t* orb_symmetries;
17     double rs_symmetries[MAX_N_SYM][3][3];
18
19     index_t n_hop;
20     rs_hopping_t* hop;
21     hamiltonian_generator_t hfill;
22
23     int SU2;
24     index_t n_spin;
25
26     index_t n_vert;
27     rs_vertex_t* vert;
28
29     tu_formfactor_t* tu_ff;
30     index_t n_tu_ff;
31
32     index_t n_vert_chan[3];
33
34     channel_vertex_generator_t vfill;
35     full_vertex_generator_t ffill;
36     greensfunc_generator_t gfill;
37     greensfunc_generator_t gproj;
38
39     char* data;
40     index_t nbytes_data;
41
42     internals_t* internals;
43 };
```

We will detail the use of all fields within the structure in the following. For initialization of the `diverge_model_t` struct, we provide the function[2]

```
1 // C/C++:
2 #include <diverge.h>
3 ...
4 diverge_model_t* model = diverge_model_init();
```

```
1 # Python:
2 import diverge
3 ...
4 model = diverge.model_init()
```

This function returns a pointer (i.e., a handle) to a `diverge_model_t` structure. It furthermore ensures a sensible default of all optional parameters, but does not set the required ones. For the destruction of the structure, the function

```
1 void diverge_model_free( diverge_model_t* model );
```

is provided[3]. The resulting skeleton that initializes the library and a model handle is given in

---

**Example 1** The skeleton of a simple example program using the divERGe C(++) library is formed by: including the function declarations (ll. 1-2), initializing the library (ll. 5-6), allocating a `diverge_model_t` structure (l. 7), and freeing resources (ll. 97-98).

---

```
1  #include <diverge.h>
2  #include <diverge_Eigen3.hpp>
3
4  int main(int argc, char** argv) {
5      diverge_init( &argc, &argv );
6      diverge_compilation_status();
7      diverge_model_t* model = diverge_model_init();
                               ...
97     diverge_model_free( model );
98     diverge_finalize();
99 }
```

---

Example 1. In the `diverge_model_t` structure, the following data fields *must* be set

- `nk[3]`: number of k-points for the vertex in direction of the reciprocal lattice vectors. If non-zero, periodicity along that direction is assumed: To simulate a 2D model we have to set `nk[2]=0`.

- `nkf[3]`: number of fine k-points around each coarse one in each direction for the loop integrals. Dimensionality has to match `nk[3]`.

- `n_orb`: number of orbitals and sites in the unit cell.

- `lattice[3][3]`: Bravais-lattice vectors in 3D. Assumes C-ordering: `lattice[0][i]` is $i$-th component of the first Bravais-lattice vector.

- `positions[MAX_N_ORBS][3]`: positions of each site and orbital — the first `n_orb` entries will be used as the positions[4].

- `SU2`: Set to `true` means that the model is $SU(2)$ symmetric and therefore spins are implicit.

- `n_spin`: number of spin quantum numbers ($n_{\text{spin}} = (S + 1/2) \cdot 2$), with the exception that for $SU(2)$ symmetric systems `n_spin` should be set to one. If the model is not $SU(2)$ symmetric the interactions are forcibly symmetrized by the crossing relations (ensuring that Pauli's principle holds).

---

[2]In general, the Python wrappers discard the `diverge_` prefix present in the C functions. We solely describe the C interface in the following, as the Python interface follows tightly.

[3]In addition to releasing internally allocated data and the handle, it takes care of releasing all user-allocated data attached to the `diverge_model_t` handle.

[4]The number of orbitals usable in divERGe is limited to `MAX_N_ORBS=32768`, which should be sufficient for most calculations, but can be changed by recompiling the library.

From C(++), setting all nonzero parameters results in Example 2. To access a field ("`field`")

---

**Example 2** Setting simple model parameters: A triangular lattice (ll. 8-12) is written directly to the arrays, whereas the basis of a honeycomb lattice (ll. 13-16) is calculated using the lattice vectors. Other required and optional parameters are set in lines 18-28.

```
                                    ...
 8    // lattice & sites
 9    model->lattice[0][0] = 1.0;
10    model->lattice[1][0] = cos(M_PI/3.);
11    model->lattice[1][1] = sin(M_PI/3.);
12    model->lattice[2][2] = 1.0;
13    Map<Mat3d> positions(model->positions[0]);
14    Map<Mat3d> latt_vecs(model->lattice[0]);
15    positions.col(0) = -1./3. * (latt_vecs.col(0) + latt_vecs.col(1));
16    positions.col(1) =  1./3. * (latt_vecs.col(0) + latt_vecs.col(1));
17    // other model parameters
18    index_t nk = 24, nkf = 15;
19    model->n_orb = 2;
20    model->n_spin = 1;
21    model->SU2 = true;
22    model->nk[0] = model->nk[1] = nk;
23    model->nkf[0] = model->nkf[1] = nkf;
24    model->n_ibz_path = 4;
25    model->ibz_path[1][0] = 0.5;
26    model->ibz_path[2][0] = 2./3.;
27    model->ibz_path[2][1] = 1./3.;
28    strcpy( model->name, "graphene" );
                                    ...
```

from Python using the handle ("`model`") returned by the routine `diverge.model_init()`, one must use `model.contents.field`. In order to facilitate the interfacing with numpy arrays, the `diverge.view_array` function is given. It returns an array view of the chunk of memory that is input as parameter. Usage is detailed in the Python examples (see the divERGe repository [18]). Beyond these simple fields, the `diverge_model_t` struct contains more complicated members, explained in greater detail in the following.

### 3.1.1 Kinetics: `rs_hopping_t`

In spirit similar to `Wannier90` [56–58], we define one single hopping parameter through the following structure:

```
1 struct rs_hopping_t {
2     index_t R[3];
3     index_t o1, o2, s1, s2;
4     complex128_t t;
5 };
```

The number of such hopping parameters given in `diverge_model_t` is `n_hop`. From C the struct is initialized using plain `malloc`, while from Python, using the provided function `diverge.alloc_array(shape, dtype)` is required in order to bypass Python's garbage collector[5].

The value of the hopping parameters can be readily read off from the kinetic part of the Hamilton operator $\hat{T}$ as the matrix elements

$$t_{o_1,o_2,s_1,s_2,\boldsymbol{R}} = \langle \boldsymbol{R}, o_2, s_2 | \hat{T} | \boldsymbol{0}, o_1, s_1 \rangle . \tag{11}$$

where `oi` are the orbitals/sites as specified by `positions` (and `n_orb`), `si` are the spin quantum numbers and `R[3]` are the shifts in Bravais lattice vectors between $o_1$ and $o_2$. Thereby, the

---

[5]Note that after having initialized the hopping array, one has to make the parameter `hop` point to the right location in memory. In the Python interface, this reads `model.contents.hop = hoppings.ctypes.data`

distance is defined as $d = r_2 + R - r_1$, if this definition is not obeyed during model creation, the default symmetry generation provided in divERGe will not work. For $SU(2)$ symmetric systems, the spin indices $s_1$ and $s_2$ are ignored in the hopping parameter structure and hence a diagonal spin sector is assumed. In practice, setting hopping parameters for a model amounts to a loop similar to the one given in Example 3.

---

**Example 3** Determining the nearest and next-nearest neighbor distance (ll. 30-31), allocation of hopping parameters (l. 33), looping over lattice vectors (l. 34) and site indices (l. 35), and setting the hopping parameters (ll. 39-42) if the corresponding length matches (ll. 37-38).

```
                                     ...
29      // hopping parameters
30      double t1_dist = positions.col(0).norm(),
31             t2_dist = latt_vecs.col(0).norm();
32      double t1 = 1.0, t2 = 0.1;
33      model->hop = (rs_hopping_t*) calloc( 1024, sizeof(rs_hopping_t) );
34      for (int Rx=-5; Rx<=5; ++Rx) for (int Ry=-5; Ry<=5; ++Ry)
35      for (int o=0; o<2; ++o) for (int p=0; p<2; ++p) {
36          assert( model->n_hop < 1024 );
37          double dist = ( latt_vecs.col(0) * Rx + latt_vecs.col(1) * Ry +
38                          positions.col(p) - positions.col(o) ).norm();
39          if (fabs(dist - t1_dist) < 1e-5)
40              model->hop[model->n_hop++] = rs_hopping_t{ {Rx,Ry,0}, o,p, 0,0,
            ↪ t1 };
41          if (fabs(dist - t2_dist) < 1e-5)
42              model->hop[model->n_hop++] = rs_hopping_t{ {Rx,Ry,0}, o,p, 0,0,
            ↪ t2 };
43      }
                                     ...
```

---

For convenient usage of divERGe as a post-processing tool for *ab-initio* simulations, we interface our code to `Wannier90` Hamiltonian files (`_hr.dat` files) with the function

```
1 rs_hopping_t* diverge_read_W90_C( const char* fname, index_t nspin,
2     index_t* len );
```

A call to `diverge_read_W90_C`[6] allocates and fills the `rs_hopping_t` array and sets `*len` to the number of elements that were read. `nspin` describes the spin quantum numbers in the `_hr.dat` file. A default value of $\texttt{nspin} = 0$ amounts to an $SU(2)$ symmetric model. If $\texttt{nspin} \neq 0$, it must be set as $|\texttt{nspin}| = 2S + 1$, with $S$ the physical spin (i.e., for $S = 1/2$ we have $|\texttt{nspin}| = 2$). The sign determines whether the spin index is the one which increases memory slowly (negative) or fast (positive) in the `_hr.dat` file. Note that the divERGe convention is always $(s, o)$[7], i.e. spin indices walking through memory faster than orbital indices ('outer indices'). In Example 3, we could — instead of explcitly looping over hopping parameters — substitute lines 30-40 with the following function call (given `graphene_hr.dat` contains hoppings):

```
1 model->hop = diverge_read_W90_C( "graphene_hr.dat", 0, &model->n_hop );
```

### 3.1.2   Interactions: `rs_vertex_t`

In general, any static two particle-interaction can be written as

$$\hat{V} = \sum_{1234} V_{1234} c_3^\dagger c_4^\dagger c_2 c_1. \tag{12}$$

Thus in all generality, we can specify the two particle interaction by its dependency on four orbitals, four spins and three momenta. This scaling in system size with power 3 (or even

---

[6]In C++, we provide the simplified `diverge_read_W90` (and in Python `diverge_read_W90_PY`) in order to not having to deal with pointers to pointers.

[7]assuming C-style ordering of arrays

4) discourages users from a direct initialization of the four point object (however it is possible, see Appendix A.4). Luckily, a fair subclass of possible interactions can be formulated as inter-orbital bilinear vertices in one of the three inequivalent interaction channels [59]. The simple interface therefore restricts the possible input to such vertices that can be efficiently represented in real space[8,9]. Similar to the hopping parameter definition (cf. Section 3.1.1), we define the structure for vertices as

```
struct rs_vertex_t {
    char chan;
    index_t R[3];
    index_t o1, o2, s1, s2, s3, s4;
    complex128_t V;
};
```

The allocation of the structure is analog to the one of the `rs_hopping_t`. It differs from the hopping parameters only in two points: The interaction channel is given as character that can either be 'C' (for crossed particle-hole, i.e., the $C$-channel), 'D' (for direct particle-hole / $D$) or 'P' (for particle-particle / $P$) and the user can supply four spin indices instead of two. In the three interaction channels, the single vertices thus represent the following terms in the interaction part $\hat{V}$ of the Hamiltonian:

$$\text{chan} = P: \quad V^{s_a,s_b,s_c,s_d}_{o_a,o_b,o_c,o_d}(\boldsymbol{R})\delta_{o_a,o_b}\delta_{o_c,o_d}\delta_{\boldsymbol{r}_{o_c}+\boldsymbol{R},\boldsymbol{r}_{o_a}} \equiv P^{s_a,s_b,s_c,s_d}_{o_a,o_c}(\boldsymbol{R}), \tag{13}$$

$$\text{chan} = C: \quad V^{s_a,s_b,s_c,s_d}_{o_a,o_b,o_c,o_d}(\boldsymbol{R})\delta_{o_a,o_d}\delta_{o_c,o_b}\delta_{\boldsymbol{r}_{o_c}+\boldsymbol{R},\boldsymbol{r}_{o_a}} \equiv C^{s_a,s_d,s_c,s_b}_{o_a,o_c}(\boldsymbol{R}), \tag{14}$$

$$\text{chan} = D: \quad V^{s_a,s_b,s_c,s_d}_{o_a,o_b,o_c,o_d}(\boldsymbol{R})\delta_{o_a,o_c}\delta_{o_d,o_b}\delta_{\boldsymbol{r}_{o_d}+\boldsymbol{R},\boldsymbol{r}_{o_a}} \equiv D^{s_a,s_c,s_d,s_b}_{o_a,o_d}(\boldsymbol{R}). \tag{15}$$

Notably, each term on the right hand side corresponds to a single `rs_vertex_t`. Note that the orbitals are initialized in a channel specific form in this interface. For a code snippet, see Example 4. Examples for different interactions, such as how to initialize a Hubbard-Kanamori interaction or a long range Coulomb interaction are given in the examples. As in the case of hopping parameters, spin indices $s_{1,...,4}$ are ignored when simulating $SU(2)$ symmetric models. Moreover, a default spin configuration can be given for non-$SU(2)$ systems as a special case: For $s_1 = -1$, the spin dependence of a given `rs_vertex_t` is initialized such that

$$I_{s_1,s_2,s_3,s_4} = \delta_{s_1,s_3}\delta_{s_2,s_4}, \tag{16}$$

where $I$ is any of the three channels. This ensures that the initialization of a non-$SU(2)$ Hubbard-Kanamori interaction is identical to the one in an $SU(2)$ system when crossing is enforced (see examples). We stress here that if a non-crossing symmetric interaction is initialized and crossing symmetry enforcement is turned off, the different backends do *not* have to give compatible results, as there is an arbitrary freedom of parametrization in the flow equations. Furthermore, the results in such a simulation are, in general, unphysical (violating Pauli's principle).

We are aware that with this interface, some two particle interactions cannot be encoded. To circumvent this shortcoming, the code offers the possibility to provide a custom vertex generation function for the full four point vertex; `full_vertex_generator_t` (cf. Appendix A.4).

## 3.2 Optional but recommended: point group symmetries

When allocating a `diverge_model_t` structure and inputting all the variables described above, your code will already run. However, we recommend to also provide divERGe with the symmetries of the model. Depending on the backend, this will reduce the runtime and memory footprint or make the results symmetry preserving [52].

---

[8]i.e., without the need to specify three real space vectors

[9]For vertices that *are* native to one interaction channel but have a natural representation in momentum space rather than real space, using a custom `channel_vertex_generator_t` can be useful (cf. Appendix A.3).

**Example 4** Setting the interaction parameters: Allocation (l. 48), setting a Hubbard-$U$ on both sites (ll. 49-50), looping over lattice vectors (l. 51) and sites (l. 52), and setting the longer ranged vertex elements (ll. 56-59) for the corresponding distance (l. 54). Note how the interaction parameters closely follow the logic for hopping parameters (cf. Example 3).

```
                                        ...
44      // interaction parameters
45      double V1_dist = positions.col(0).norm(),
46             V2_dist = latt_vecs.col(0).norm();
47      double V0 = 3.6, V1 = 0.1, V2 = 0.05;
48      model->vert = (rs_vertex_t*) calloc( 1024, sizeof(rs_vertex_t) );
49      model->vert[model->n_vert++] = rs_vertex_t{ 'D', {0,0,0}, 0,0, -1,0,0,0,
            ↪ V0 };
50      model->vert[model->n_vert++] = rs_vertex_t{ 'D', {0,0,0}, 1,1, -1,0,0,0,
            ↪ V0 };
51      for (int Rx=-5; Rx<=5; ++Rx) for (int Ry=-5; Ry<=5; ++Ry)
52      for (int o=0; o<2; ++o) for (int p=0; p<2; ++p) {
53          assert( model->n_vert < 1024 );
54          double dist = ( latt_vecs.col(0) * Rx + latt_vecs.col(1) * Ry +
55                          positions.col(p) - positions.col(o) ).norm();
56          if (fabs(dist - V1_dist) < 1e-5)
57              model->vert[model->n_vert++] = rs_vertex_t{ 'D', {Rx,Ry,0}, o,p,
            ↪ -1,0,0,0, V1 };
58          if (fabs(dist - V2_dist) < 1e-5)
59              model->vert[model->n_vert++] = rs_vertex_t{ 'D', {Rx,Ry,0}, o,p,
            ↪ -1,0,0,0, V2 };
60      }
                                        ...
```

The point-group symmetries of the model can be attached to a handle (`diverge_model_t`) via two arrays and their length (cf. ls. $15 - 17$ in the listing on p. 5), i.e.,

```
1      index_t n_sym;
2      complex128_t* orb_symmetries; // (n_sym, n_spin*n_orb, n_spin*n_orb)
3      double rs_symmetries[MAX_N_SYM][3][3]; // (n_sym,3,3)
```

where `rs_symmetries` stores the real-space transformations ($3 \times 3$ matrices $\mathbf{M}$, with $r' = \mathbf{M}r$) and `orb_symmetries` stores the transformation behavior of the combined state of spin and orbit. `n_sym` is the number of symmetries present in the model. The symmetries can be directly provided by the user, however this can become cumbersome for multi-orbital and/or multi-site models. Hence we provide

```
1 void diverge_generate_symm_trafo( index_t n_spin,
2    const site_descr_t* orbs, index_t n_orbs, const sym_op_t* syms,
3    index_t n_syms, double* rs_trafo, complex128_t* orb_trafo );
```

to generate both the real space transformation (`rs_trafo`) and the orbital/sublattice/spin space transformation (`orb_trafo`) for a symmetry operation specified by the list of symmetry operators stored in `syms` and for orbitals specified by the site descriptors stored in `orbs`. We will explain these two structs in the following. First, a single symmetry operator is defined by the structure

```
1 typedef struct sym_op_t {
2     char type;
3     double normal_vector[3];
4     double angle;
5 } sym_op_t;
```

where `type` specifies the type of symmetry, with possible values 'R' (rotation), 'M' (mirror), 'I' (inversion), 'S' (spin rotation), 'F' (spin flip), and 'E' (identity). `normal_vector[3]` encodes the normal vector to the plane in which the symmetry operation acts and `angle` the angle of the rotation in degrees (ignored if the operation does not need an angle). To encode, for example, a mirror in the $yz$-plane, we can set the following for the structure's fields:

```
1      sym_op_t op;
```

```
2      op.type = 'M';
3      op.normal_vector[0] = 1.0;
4      op.normal_vector[1] = 0.0;
5      op.normal_vector[2] = 0.0;
6      op.angle = 0.0;
```

A rotation by 90° around the $z$ axis reads

```
1      sym_op_t op;
2      op.type = 'R';
3      op.normal_vector[0] = 0.0;
4      op.normal_vector[1] = 0.0;
5      op.normal_vector[2] = 1.0;
6      op.angle = 90.0;
```

In the call to `diverge_generate_symm_trafo`, the parameter `n_syms` specifies the number of elementary symmetry operations needed to describe the current symmetry. The order of application of these elementary operations $O_i$ to some vector $v$ is $O_{n-1}[O_{n-2}[\ldots O_1[O_0[v]]]]$.

As the symmetry generator also works for tight-binding orbitals with non-trivial angular dependence, we require the `site_descr_t` structure, which describes the content of real spherical harmonics of the tight-binding basis function for each orbital- and site index: `orbs` and the number of elements `n_orbs`[10]. The structure is defined as

```
1 typedef struct site_descr_t {
2      complex128_t amplitude[MAX_ORBS_PER_SITE];
3      real_harmonics_t function[MAX_ORBS_PER_SITE];
4      index_t n_functions;
5      double xaxis[MAX_ORBS_PER_SITE][3];
6      double zaxis[MAX_ORBS_PER_SITE][3];
7 } site_descr_t;
```

Again the interface is inspired by the orbital interface of `Wannier90` [56–58]. `n_functions` gives the number of basis functions required for constructing the orbital and `functions` gives the type of spherical harmonic, while `amplitude` is the complex weight of each of the basis functions. For example, to generate a $p_+$ orbital, we require a $p_x$ and a $p_y$ orbital with weights of $1/\sqrt{2}$ and $i/\sqrt{2}$ respectively. In case the coordinate system of the real spherical harmonic does not align with the coordinate system of the lattice, we also allow for the arguments `xaxis` and `zaxis` which are used to define the x and z axis of the real harmonic, these are optional and, if not changed by the user, the two coordinate systems are assumed to align. The `functions` argument is set via enumeration values `real_harmonics_t` (see Appendix B.2). The examples include models of various symmetry using both the Python and the C/C++ interfaces.

### 3.3  Preparing the Model for a Flow

After the model struct is initialized and filled with user data, internal structures must be generated. In addition to common internal structures, each backend defines its own internal structure. Since users may want to do something with the Hamiltonian array, the energies, the momentum meshes, etc. (*common* internal structures), the common internal structures are initialized separately from the backend specific ones. Before starting with these potentially computationally expensive operations, it is advised to check for obvious errors in the model. These two steps result in the following lines of code:

```
1 if (diverge_model_validate( model ))
2      printf("something went wrong!\n");
3 diverge_model_internals_common( model );
```

---

[10]We stress here that the value `n_orbs` passed to the symmetry generation routine does not have to equal `n_orb` from the model, but only should be if you are planning to use the resulting symmetry transformations *directly* in the model.

After the validity check and initialization of common internals, many codes will in practice call one of the following functions:

```
1 void diverge_model_internals_grid( diverge_model_t* m );
2 void diverge_model_internals_patch( diverge_model_t* m, index_t np_ibz );
3 void diverge_model_internals_tu( diverge_model_t* m, double maxdist );
```

which initialize the backend specific internals for the grid (l. 1), $N$-patch (l. 2) or the TUFRG (l. 3) backend. Be aware that in the case of $N$-patch FRG, a Fermi surface patching is generated automatically[11]. This renders the above function sensitive to the model's chemical potential $\mu$. Users are thus expected to adjust the value of $\mu$ (see Section 3.4) *before* the backend specific internals are set, but *after* the common ones, since the call to

```
1 void diverge_model_internals_common( diverge_model_t* m );
```

contains the diagonalization of the Hamiltonian. This also implies that the Hamiltonian cannot be changed after calling this function[12]. Changing the default Hamiltonian generated by diverge is easiest accomplished by writing a custom Hamiltonian generator that, as a first step, calls the default Hamiltonian generator (see Appendix A.5).

## 3.4 Some Convenience Functions

Often times, one wishes to perform simulations of a model at several values of chemical potential $\mu$ or filling $\nu$ (where in our convention, $\nu = 0$ corresponds to a completely empty model and $\nu = 1$ to a completely filled one). Three convenience functions are defined in divERGe to perform tasks related to changing $\mu$ or $\nu$ of a given model:

```
1 double diverge_model_get_filling( diverge_model_t* model,
2     const double* E, index_t nb );
3 double diverge_model_set_filling( diverge_model_t* model,
4     double* E, index_t nb, double nu );
5 void diverge_model_set_chempot( diverge_model_t* model,
6     double* E, index_t nb, double mu );
```

For all of them, the energy buffer (E) may be NULL (or None from Python), allowing for usage of the *internally* constructed energy arrays (and number of bands) instead of E and nb. The function `diverge_model_set_filling`, aside from setting $\mu$ correctly, returns the value $\mu$ that was needed to fill the model to $\nu$ (at $T = 0$).

Furthermore, some of the internal structures may be accessed from outside through

```
1 double* diverge_model_internals_get_E( const diverge_model_t* model );
2 complex128_t* diverge_model_internals_get_U( const diverge_model_t* model );
3 complex128_t* diverge_model_internals_get_H( const diverge_model_t* model );
4 double* diverge_model_internals_get_kmesh( const diverge_model_t* model );
5 double* diverge_model_internals_get_kfmesh( const diverge_model_t* model );
```

Note that in advanced use cases (when, e.g., the standard Hamiltonian generator is overwritten), the first three of those functions are not guaranteed to return meaningful results.

Lastly, the user can access point group symmetrization routines for use on arbitrary arrays of certain shape and data type:

```
1 double diverge_symmetrize_2pt_coarse( diverge_model_t* model,
2     complex128_t* buf, complex128_t* aux );
3 double diverge_symmetrize_2pt_fine( diverge_model_t* model,
4     complex128_t* buf, complex128_t* aux );
5 double diverge_symmetrize_mom_coarse( diverge_model_t* model,
6     double* buf, index_t sub, double* aux );
7 double diverge_symmetrize_mom_fine( diverge_model_t* model,
8     double* buf, index_t sub, double* aux );
```

---

[11]This behavior can be changed by manually generating the patching structure before calling the function `diverge_model_internals_patch`, see Appendix A.1.

[12]Under the assumption of using the default Green's function generator.

```python
#!/usr/bin/env python3
#coding:utf8
import diverge.output as do
import matplotlib.pyplot as plt

M = do.read('model.dvg')

fig = plt.figure(layout="constrained", figsize=(3,2))
xvals = do.bandstructure_xvals(M)
plt.plot( xvals, do.bandstructure_bands(M), c='navy' )
plt.xticks(do.bandstructure_ticks(M), [r'$\Gamma$',r'$M$',r'$X$',r'$\Gamma$'])
plt.xlim( xvals[0], xvals[-1] )
plt.ylabel( r'$\epsilon_b({\bf k})$' )
fig.savefig('bands.pdf')
```

Listing 1: Simple Python script to plot the band structure of a divERGe model file saved in `model.dvg`. The resulting plot, under the assumption of a three-band Emery model, is shown in Fig. 1.

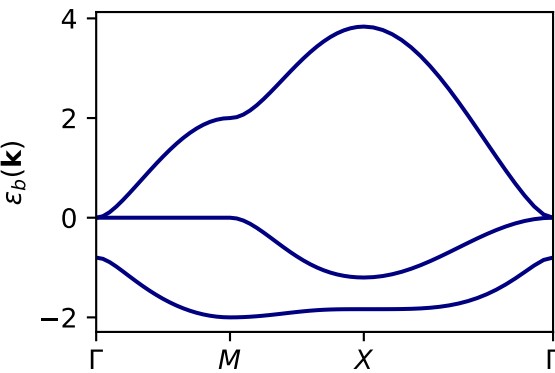

Figure 1: Band structure of a three-band Emery model plotted with the script given in Listing 1. The model parameters (and the file `model.dvg`) are those from the Emery example: `examples/python_tutorial/emergent-3-model-details.py` in the git repository of divERGe [18] (or Appendix E.1).

The `_fine` (`_coarse`) functions act on arrays where the momentum dimension is equivalent to the fine (coarse) momentum mesh. In case the auxiliary buffers (`aux`) are provided, they must be of the same shape as the main buffer (`buf`). If they are NULL, internal allocations are performed. We offer symmetrization of 2-point functions [`_2pt`, shape $(n_k, n_b, n_b)$], and diagonal functions [`_mom`, e.g. energy arrays, shape $(n_k, \text{sub})$].

## 3.5 Model Output

Before performing the flow (see Section 4), which usually presents the computationally most expensive task, it is advised to check whether the model is implemented correctly. Notably, this step is often computationally cheap enough to be performed on a local machine. We allow for models to be written to disk via

```c
char* diverge_model_to_file( diverge_model_t* m, const char* name );
char* diverge_model_to_file_finegrained( diverge_model_t* m, const char* name,
    const diverge_model_output_conf_t* cfg );
```

Using the second function, users can achieve fine-grained control over optional output such as the momentum meshes, or the dispersion and orbital-to-band matrices in the primitive zone. As return value, the above functions give the MD5 checksum of the output file as a string. Details on the binary file format and optional output parameters are given in Appendix C.1.

As part of divERGe, we ship the Python library `diverge.output` that reads all divERGe output files, including the model file and postprocessing files (see Section 5). Plotting the band structure of a model merely requires a few lines of Python: Listing 1 contains all the code that produces the band structure plot of a three-band Emery model shown in Fig. 1.

The steps described in Sections 3.3 to 3.5 are practically illustrated as a code snippet in Example 5.

---

**Example 5** Preparing a model for the flow: Validation (l. 62) precedes setting the common internals (l. 67), the filling (l. 68), and the TUFRG specific internals (l. 69). We also showcase model output to a file setting some of the non-default parameters (ll. 70-74).

```
                                    ...
61    // check!
62    if (diverge_model_validate(model))
63        diverge_mpi_exit(EXIT_FAILURE);
64    // finalize model and save to disk
65    double filling = 0.6,
66           ffdist = 2.1;
67    diverge_model_internals_common( model );
68    diverge_model_set_filling( model, NULL, -1, filling );
69    diverge_model_internals_tu( model, ffdist );
70    diverge_model_output_conf_t cfg =
          ↪ diverge_model_output_conf_defaults_CPP();
71    cfg.E = true;
72    cfg.npath = -1;
73    cfg.kf_ibz_path = 1;
74    diverge_model_to_file_finegrained( model, "graphene_model.dvg", &cfg );
                                    ...
```

# 4 Performing the Flow

## 4.1 A single flow step

Given that the user has allocated, filled, and initialized (the internals of) a `diverge_model_t` structure, the next step towards performing an FRG flow is through the opaque structure `diverge_flow_step_t` (serving as a handle). Allocation of which requires a fully initialized `diverge_model_t` structure and must be performed via one of the following two functions:

```
1 diverge_flow_step_t* diverge_flow_step_init( diverge_model_t* model,
2     const char* mode, const char* channels );
3 diverge_flow_step_t* diverge_flow_step_init_any( diverge_model_t* model,
4     const char* channels );
```

The second function is valid if and only if a single set of internal structures (corresponding to a single backend) has been initialized in the model. As users may want to initialize multiple backends, we provide the first of the two functions for precise control via the `mode` parameter (that can be `"grid"`, `"patch"`, or `"tu"`). In both cases, `channels` is passed as string and encodes which interaction channels are included in the FRG flow. If the character 'P' ('C', 'D') is found in the `channels` string, the respective diagrams are calculated, allowing easy access to RPA calculations in any of the interaction channels. For systems without $SU(2)$ symmetry, one must not include the $D$-channel without the $C$-channel and vice-versa. Otherwise, Graßmann symmetry would be broken. In addition to the interaction channels, the character 'S' stands for the inclusion of (static) self-energies. Currently, these are only supported in the TUFRG backend, which is subject to change in future releases.

After the initialization, the `diverge_flow_step_t` handle serves the purpose of performing Euler integration steps of the FRG flow [Eqs. (2) to (5)]. One integration step starting at $\Lambda$ and going to $\Lambda + d\Lambda$ is calculated by calling

```
1 void diverge_flow_step_euler( diverge_flow_step_t* step, double Lambda,
2     double dLambda );
```

We stress that *negative* $d\Lambda$ is required to flow from high $\Lambda$ to zero. Since users often wish to loop over flow steps, we provide a simple adaptive integrator. Its usage is explained below.

## 4.2 Integrating the flow equations

Under the approximations and assumptions taken within the scope of divERGe, the flow equations are integrated from high scales $\Lambda = \infty$ to low scales $\Lambda = 0$ until a phase transition (divergence of a vertex element) is encountered or a minimal $\Lambda$ is hit. In practice we "flow" using a construction similar to loop shown in Example 6, where the function call to `diverge_euler_defaults_CPP()` returns sensible defaults for many models (cf. Appendix B.1). This code snippet will integrate the flow equations until a stopping criterion

---

**Example 6** Flow step initialization (l. 78) and cleanup (l. 96), integrator setup (the default values are reasonable for most models; we change them to showcase the mechanism; ll. 79-82), and flow loop idiom (ll. 83-88).

```
                                        ...
75    // flow
76    double vmax = 0;
77    double cmax[3] = {0};
78    diverge_flow_step_t* step = diverge_flow_step_init_any( model, "PCD" );
79    diverge_euler_t eu = diverge_euler_defaults_CPP();
80    eu.Lambda = 10.0;
81    eu.dLambda = -1.0;
82    eu.maxvert = 10.0;
83    do {
84        diverge_flow_step_euler( step, eu.Lambda, eu.dLambda );
85        diverge_flow_step_vertmax( step, &vmax );
86        diverge_flow_step_chanmax( step, cmax );
87        mpi_printf( "%.5e %.5e %.5e %.5e %.5e\n", eu.Lambda, cmax[0],
            ↪ cmax[1], cmax[2], vmax );
88    } while (diverge_euler_next( &eu, vmax ));
                                        ...
96    diverge_flow_step_free( step );
                                        ...
```

---

is met[13]. After each integration step the snippet outputs the channel and vertex maxima, as well as the current value of the integration scale $\Lambda$. The step-width is adjusted to keep the error tangible while at the same time being efficient (cf. Appendix B.1). Note that when selecting only a specific channel in the flow step initialization, the FRG flow amounts to an RPA calculation in that channel (with formfactors accounting for non-channel-specific long range interactions).

We specifically chose to leave control to the user regarding the flow loop, as it is never performance critical and there are many things that can be done at each step of performing the flow. For example, vertices may be accessed through `diverge_flow_step_vertex`. The indices etc. of the returned array are of course backend dependent. Nonetheless, one can use it to, e.g., track specific components of the vertex. If the selfenergy is included, we additionally provide functions to adjust the filling after each step by calling `diverge_flow_step_refill`, leading to a quasi-canonical description. Note that this quasi-canonical description may be ill-defined in many cases and is therefore not enabled by default. For further usage see the online documentation[14] as well as the examples.

## 5 Post Processing and Output

After integration of the flow equations and reaching a stopping criterion of the FRG flow, susceptibilities, vertex eigenvectors or solutions to linearized gap equations may provide physical insights to the system [20, 28, 40, 51, 52]. Depending on the backend, the functions

---

[13]usually either a channel becoming larger than a predefined value or a minimal scale

[14]https://frg.pages.rwth-aachen.de/diverge/

```python
#!/usr/bin/env python3
#coding:utf8
import diverge.output as do
import matplotlib.pyplot as plt
import numpy as np

# helper function to map a complex array to colors
def complex_cmap( ary ):
    A = ary.flatten()
    C = plt.cm.hsv( np.angle(A)/(2*np.pi)+0.5)
    C[:,3] *= np.abs(A) / np.abs(A).max()
    return C.reshape((*ary.shape, 4))

M = do.read('honeycomb.cpp.mod.dvg')
O = do.read('honeycomb.cpp.out.dvg')

K = O.P_qV[0][1]
vector_colors = complex_cmap( O.P_qV[0][3][1:3] )
# select eigenvectors [1:3] and map to colors

fig, axs = plt.subplots( 2, 4, sharex=True, sharey=True,
                         layout="constrained", figsize=(3,3) )
for idx in range(2):
 for o1 in range(2):
  for o2 in range(2):
    ax = axs[o1,idx*2+o2]
    ax.scatter( *M.kmesh[K,:2].T, c=vector_colors[idx,:,o1,o2], s=70, lw=0 )
    ax.set_aspect(1); ax.set_xticks([]); ax.set_yticks([])
    ax.set_title( r'$\Delta^{%i}_{%i%i}({\bf k})$' % (idx+1,o1+1,o2+1),
                 fontsize=9 )
fig.savefig( 'patch.pdf' )
```

Listing 2: Python code to produce Figure 2 from the output generated by `examples/c_cpp/honeycomb.cpp` [18]. As the vertex eigenvectors for the honeycomb lattice Hubbard model are complex in general, we must declare a helper function to plot these complex arrays (`complex_cmap`). We select the eigenvectors at indices 1 and 2; those are the ones corresponding to $d_{xy}$- and $d_{x^2-y^2}$-wave superconducting gaps.

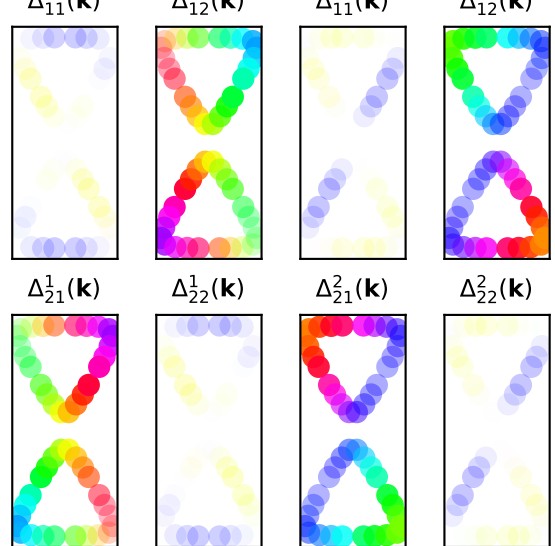

Figure 2: Visualization of the $d_{xy}$ and $d_{x^2-y^2}$ superconducting gap functions of the honeycomb lattice Hubbard model calculated in $N$-patch FRG (`examples/c_cpp/honeycomb.cpp` [18] or Appendix E.2). The plot is generated with the small Python script presented in Listing 2. We encode the complex phase as color, and the magnitude as opacity. Notice how most of the orbital weight of the gap function is on the off-diagonal elements $\Delta_{12}(\boldsymbol{k})$ and $\Delta_{21}(\boldsymbol{k})$.

```c
void diverge_postprocess_and_write( diverge_flow_step_t* s,
    const char* name );
void diverge_postprocess_and_write_finegrained( diverge_flow_step_t* s,
    const char* name, const diverge_postprocess_conf_t* cfg );
```

perform an optimized set of these post-processing tasks and write the results to disk. Note that we take the same approach as for `diverge_model_t` (cf. Section 3.5) in how the default behavior can be changed by calling the fine-grained function with an additional configuration structure. Details on the available parameters and file formats are found in Appendix C.2, a practical example is given in Example 7.

---

**Example 7** Doing post-processing given a flow step instance. The default parameters (l. 90) can be changed (ll. 91-93) and passed to the post-processing routine (l. 94). The code obtained from merging the snippets given in Examples 1 to 7 can be found in `examples/c_cpp/honeycomb_tutorial.cpp` [18].

```
                                           ...
89    // post processing
90    diverge_postprocess_conf_t out = diverge_postprocess_conf_defaults_CPP();
91    out.tu_storing_relative = true;
92    out.tu_storing_threshold = 0.8;
93    out.tu_which_solver_mode = 's';
94    diverge_postprocess_and_write_finegrained( step, "graphene_post.dvg",
          ↪ &out );
                                           ...
```

---

All post-processing and model (cf. Section 3.5) files can be read into Python classes using the library `diverge.output`. The file type is automatically recognized. As a demonstration, we plot the (negative) leading $P$-channel eigenvectors of a honeycomb lattice Hubbard model close to van Hove doping (the parameters are equivalent to those given in Fig. 4 (b) of Ref. [52], at filling $v = 0.6$). The few lines of Python in Listing 2 (mostly matplotlib) result in the visual representation of the two degenerate superconducting $d_{xy}$- and $d_{x^2-y^2}$-wave states shown in Fig. 2.

# 6 Conclusion

In this paper, we presented in detail the usage of the divERGe library. We focused on explaining the interface for in- and output as well as the backend implementations. For applications of this framework on physical systems, see Refs. [**?**, 19, 20, 23, 26, 48, 51, 60]. We believe that the flavors of FRG realized in divERGe represent a sophisticated drop-in replacement for RPA in many applications in which qualitative predictions of correlated phases are wanted. As such, the divERGe library has a promising future as an extension of the *ab-initio* pipeline. We believe that the library presented here significantly ameliorates the usability of FRG as a method to study competing orders in solid state systems and hence massively increases the reach and popularity of FRG in the scope of *ab-initio* as well as model calculations. In the future, we hope that the tight connection of divERGe to Wannier90 allows to extend high-throughput materials databases [61–63] to FRG, paving the way for systematic characterization of competing electronic orders in quantum materials. Moreover we plan to further entangle divERGe with the existing ecosystem by, e.g., providing wrappers for interaction parameters obtained from first principle codes [64].

The publication of this framework can however only be seen as a first step and many possible future extensions are imaginable: First and foremost, the code in its current form does not treat the frequency dependence, which would allow us to introduce retardation effects on the two-particle level. This will not only require to "just implement" the frequency dependence, but smarter representations of the frequency content of the high-dimensional vertex functions have to be found. First steps in this direction have been recently taken [65–68]. Once this has been achieved, the usage as a post-DMFT tool becomes available through the DMF$^2$RG route

and related proposals [69–72]. Secondly, while a significant amount of time has been spent with the optimization of the backends, reducing computational effort remains a continuous challenge we strive to address.

## Acknowledgements

The authors thank J. C. L. Beyer, M. Bunney, A. Fischer and D. Rohe for helpful discussions and testing the code. JBP and LK also thank C. Honerkamp for introducing them to FRG and for all the insightful discussions on this topic.

**Funding information**     The authors gratefully acknowledge computing time granted through JARA on the supercomputer JURECA [73] at Forschungszentrum Jülich. JBP and DMK are supported by the Deutsche Forschungsgemeinschaft (DFG, German Research Foundation) under RTG 1995, within the Priority Program SPP 2244 "2DMP" — 443273985 and under Germany's Excellence Strategy - Cluster of Excellence Matter and Light for Quantum Computing (ML4Q) EXC 2004/1 - 390534769. LK gratefully acknowledges support from the DFG through FOR 5249 (QUAST, Project No. 449872909, TP5).

# A Advanced Initialization

The following fields of the `diverge_model_t` are not required to be filled by the user. In fact, we highly recommend to only use these if you know what you are doing. No guarantees for correctness of the results can be given anymore as some of these options override key functions.

## A.1 Fermi Surfaces: `mom_patching_t`

The *N*-patch FRG backend of divERGe relies on the assumption that the model is defined on a fixed momentum space given by `nk` (chosen to sufficient accuracy). Patching of the Fermi surface as well as momentum integration of the loops happens *on this momentum grid*. We can therefore guarantee, e.g., general operation for nontrivial Hamiltonians or usage with Green's functions instead of Hamiltonians. In practice, the requirement of a fixed momentum mesh simplifies many aspects of an *N*-patch FRG calculation. For example, the structure to define a patching is as simple as follows:

```
struct mom_patching_t {
    index_t n_patches;
    index_t* patches;
    double* weights;

    index_t* p_count;
    index_t* p_displ;

    index_t* p_map;
    double* p_weights;
};
```

It includes the total number of patches (`n_patches`), the indices of these patches referencing the coarse momentum mesh (`patches`), the weights assigned to each of the patches for Brillouin zone integrals (`weights`), and a detailed description of the refinement for each of the patches (`p_count, p_displ, p_map, p_weights`)[15].

In the codebase, several convenience functions that simplify usage of the `mom_patching_t` struct are defined. For example, Fermi surfaces can be found automatically using the following function:

```
void diverge_patching_find_fs_pts_C( diverge_model_t* m, double* E,
    index_t nb, index_t n_pts_ibz, index_t n_pts_search, index_t** fs_pts_ptr,
    index_t* n_fs_pts_ptr );
```

Notice how the routine operates on *arbitrary* energies and number of bands[16]. The resulting vector is allocated and written to `*fs_pts_ptr`, with its size saved in `*n_fs_pts_ptr`[17]. To generate the patching struct from the Fermi surface indices (which can of course be modified before the struct generation), we provide

```
mom_patching_t* diverge_patching_from_indices( diverge_model_t* m,
    const index_t* fs_pts, index_t n_fs_pts );
```

In addition, automatically refining the integration regions and re-symmetrizing those refined integration meshes is available via

---

[15]The first two arrays, `p_count` and `p_displ`, serve as descriptors for the third and fourth array: For a given patch index `p`, the slice `p_displ[p]:p_displ[p]+p_count[p]` of the arrays `p_map` and `p_weights` describes all indices corresponding to the refinement of patch `p` as well as their weights, respectively.

[16]Similar to other divERGe routines, passing NULL for the energy array makes the library operate on the model internals.

[17]A wrapper for C++ returning an `std::vector<index_t>` instead of allocating and filling an array at `*fs_pts_ptr` and setting its length at `*n_fs_pts_ptr` as well as a wrapper for Python returning a numpy array is defined due to the inconvenient way of dealing with pass-by-pointer.

```
1 void diverge_patching_autofine( diverge_model_t* mod, mom_patching_t* patch,
2     const double* E, index_t nb, index_t ngroups, double alpha, double beta,
3     double gamma );
4 void diverge_patching_symmetrize_refinement( diverge_model_t* mod,
5     mom_patching_t* patch );
```

We advise users to consult the documentation and/or source code in case they wish to use those functions.

To automate the procedure of generating a patching with the energies calculated from the hopping parameters supplied, the default examples leave `model->patching` untouched and instead only call

```
1 void diverge_model_internals_patch( diverge_model_t* m, index_t np_ibz );
```

Note that this call requires the common internal structures (cf. Section 3.3) to be set.

## A.2   Formfactor Expansions: `tu_formfactor_t`

The `tu_formfactor_t` structure is filled automatically when calling

```
1 void diverge_model_internals_tu(diverge_model_t* mod, double dist);
```

This call ensures that all form-factors for each site/orbital on the lattice that have a length smaller than `dist` are included. Precisely, global and local point-group symmetries are respected by this type of truncated unity [23]. The structure that is filled is exposed to the user and reads

```
1 struct tu_formfactor_t {
2     index_t R[3];
3     index_t ofrom, oto;
4     double d;
5     index_t ffidx;
6 };
7 index_t n_tu_ff;
```

where `R[3]` is the Bravais-lattice vector[18] corresponding to the form-factor. The bond connects the orbital/site indexed by `ofrom` with the one indexed by `oto` which is located in the unit cell shifted by `R[3]`. `d` is the absolute distance between `ofrom` and `oto+R[3]`. `ffidx` is always constructed by the code itself, it enumerates all possible exponentials generated by the different form-factors, which usually are far less than the form-factors themselves.

For fine-grained control over form-factors, the generation step can be bypassed by setting a nonzero value for `n_tu_ff`. This transfers responsibility of allocating and filling the `tu_formfactor_t` struct to the user. The code will then only sort it into the standardized form and set `ffidx`. As minimal requirement, each site/orbital must possess the on-site form-factor, i.e. ($[0,0,0]$,$o_1$,$o_1$,0.0,0).

## A.3   Custom Channel Based Vertex Generation

To allow for user-defined channel based vertex input, the model can be equipped with a function pointer that will be called instead of the default channel generation. The pointer is expected to have the form

```
1 int (*channel_vertex_generator_t)(diverge_model_t* model, char channel,
2     complex128_t* buf);
```

the return value tells whether the buffer has been touched (1) or not (0). If it has been touched, it is expected to contain the corresponding vertex channel in the order (nk, n_spin, n_spin, n_orb, n_spin, n_spin, n_orb). Note that this function does *not* offer more versatility than the default vertex generator, but can be used to implement interaction profiles that have a natural representation in *momentum space*, rather than real space.

---

[18]Given in units of the lattice vectors.

### A.4 Custom Full Vertex Generation

In cases where the user requires more versatility for the vertex initialization, they can attach a full vertex generator function of the form

```
void (*full_vertex_generator_t)(const diverge_model_t* model,
        index_t k1, index_t k2, index_t k3, complex128_t* buf);
```

to the model structure. The function is expected to return the full vertex at a specific momentum combination in `buf`, as the full two-particle interaction is in general far too large to store. We expect the user to not make use of parallelism within their full vertex generator, as the function is called in parallel by divERGe. The index order is expected to be (n_spin, n_orb, n_spin, n_orb, n_spin, n_orb, n_spin, n_orb). We do not encourage users to employ a custom full vertex generator.

### A.5 Custom Hamiltonian Generation

The custom Hamiltonian generator can be useful if, e.g., the Hamiltonian is present as data rather than hopping elements. It must be of the form

```
void (*hamiltonian_generator_t)(const diverge_model_t* model,
        complex128_t* buf);
```

Upon return, the Hamiltonian is stored in the buffer assuming an ordering of (prod(nk*nkf), n_spin,n_orb, n_spin,n_orb). Another use case of a custom Hamiltonian generator is modifying the Hamiltonian that is generated using the default Hamiltonian generator. In this case, one would write a function that calls the default generator first, i.e.,

```
void custom_hamilton_generator( const diverge_model_t* model,
    complex128_t* buf ) {
    diverge_hamilton_generator_default( model, buf );
    // do something with model and buf, or any global variable
}
```

and attach it to the model handle as custom Hamiltonian generator `hfill`.

### A.6 Custom Green's Function Generation

Analogously, the Green's function generator is of the structure

```
greensfunc_op_t (*greensfunc_generator_t)(const diverge_model_t* model,
        complex128_t Lambda, gf_complex_t* buf);
```

The function is expected to provide the Greens function at $\Lambda$ and $\Lambda^*$ in `buf` in the index order ($\pm\Lambda$, prod(nk*nkf), n_spin,n_orb, n_spin,n_orb). If `gfill` is set the Greens function is generated with the user defined greens-function generator, which can be useful for simulation of models where only a subspace of orbitals is correlated[19].

In addition to changing `gfill`, the user may also set another Green's function generator, `gproj`. If this one is set, divERGe calculates $L = L(G) - L(G_{\text{proj}})$. This functionality is used to remove certain subspaces from the kinetics[20]. It can also be used to isolate the effects of individual bands in the calculation. We note that when using a sharp frequency cutoff, we formally replace the regulator $f(\Lambda)$ by

$$G^{\Lambda} = G_0^{\text{proj}} + (G_0 - G_0^{\text{proj}})\Theta(|\omega| - \Lambda), \tag{17}$$

where $G_0^{\text{proj}} \equiv T$ is the propagator in the target space and $G_0 - G_0^{\text{proj}} \equiv R$ the propagator in the remote space (everything except the target space). This choice of the cutoff restricts the two

---

[19] example found in `examples/c_cpp/t2g_subspace.cpp` [18]
[20] example found in `examples/c_cpp/t2g_cfrg.cpp` [18]

particle propagator to the high energy sector while the Greens function is not restricted [59, 74]. One can easily prove that with this regulator, terms of the form

$$
\begin{aligned}
\frac{d}{d\Lambda}\big[G^\Lambda G^\Lambda\big] = \frac{d}{d\Lambda}\big[TT + RR\,\Theta^2(\Lambda) + 2TR\,\Theta(\Lambda)\big] = \\
= RR\,2\,\Theta(\Lambda)\delta(\Lambda) + 2\,TR\,\delta(\Lambda) = \big[RR + 2TR + TT - TT\big]\delta(\Lambda) = \\
= \frac{d}{d\Lambda}\big[G_0^\Lambda G_0^\Lambda\big] - \frac{d}{d\Lambda}\big[G_0^{\mathrm{proj},\Lambda} G_0^{\mathrm{proj},\Lambda}\big] \quad (18)
\end{aligned}
$$

are contained in the FRG flow, where we choose to implement the last expression for in-code simplicity.

# B Additional structures

## B.1 Controlling the flow: `diverge_euler_t`

We include an adaptive Euler integrator in the library that allows fine-grained control over the integration of the flow equations. In practice, the flow loop will often be given as

```
diverge_euler_t euler = diverge_euler_defaults; // or change the defaults
double vmax = 0.0;
do {
    diverge_flow_step_euler( step, euler.Lambda, euler.dLambda );
    diverge_flow_step_vmax( step, &vmax );
    // do something with the vertices here
} while (diverge_euler_next( &euler, vmax ));
```

Notice how the next $\Lambda$ and $d\Lambda$ are generated using the function `diverge_euler_next` within the condition of the `while` loop: It returns 1 if a next step should be performed, and 0 if the flow should be terminated. We offer control over the termination conditions and integrator through the `diverge_euler_t` structure as follows:

```
struct diverge_euler_t {
    double Lambda;
    double dLambda;
    double Lambda_min;
    double dLambda_min;
    double dLambda_fac;
    double dLambda_fac_scale;
    double maxvert;
    double maxvert_hard_limit;
    index_t niter;
    index_t maxiter;
    index_t consider_maxvert_iter_start;
    double consider_maxvert_lambda;
};
```

The meaning of each of the parameters is explained in the following:

- `Lambda`: starting scale $\Lambda$ of the flow — should be larger than the bandwidth (default: $\Lambda = 50$)

- `dLambda`: starting step-width $d\Lambda$ of the flow, has to be negative. A rough estimate is given by $d\Lambda \approx -0.1\Lambda$ (default: $d\Lambda = -5$)

- `Lambda_min`: stopping value $\Lambda_{\min}$ of $\Lambda$ if no divergence is hit — has to be nonzero due to the finite energy resolution of the simulation (default: $\Lambda_{\min} = 10^{-5}$)

- `dLambda_min`: minimal allowed step-width $d\Lambda_{\min}$ serving as a lower bound $b$ on the step-width (default: $d\Lambda_{\min} = 10^{-6}$)

- dLambda_fac: defines an upper bound $B$ for the step-width as $B_{\mathrm{fac}} = \mathrm{d}\Lambda_{\mathrm{fac}} \cdot \Lambda$ (default: $\mathrm{d}\Lambda_{\mathrm{fac}} = 0.1$)

- dLambda_fac_scale: additional scaling factor $\mathrm{d}\Lambda_{\mathrm{fac-sc}}$ for the calculation of the width of the next step, used as an upper bound $B$ for step-width: $B_{\mathrm{fac-sc}} = \mathrm{d}\Lambda_{\mathrm{fac-sc}}/V_{\max} \cdot \Lambda$ (default: $\mathrm{d}\Lambda_{\mathrm{fac-sc}} = 1.0$)

- maxvert: stopping condition such that the flow is halted when $V_{\max} > $ maxvert, i.e., the maximal element of the vertex reaches maxvert (default: maxvert $= 50$)

- maxvert_hard_limit: Hard limit for the stopping condition — especially important if consider_maxvert_iter_start is set (default: $10^4$)

- niter: current number of performed flow steps (gets updated with each call to the step-width function diverge_euler_next)

- maxiter: maximal number of performed flow steps (default: $-1$, thus ignored)

- consider_maxvert_iter_start: number of steps for which the stopping condition is ignored, usefull when starting with long range interactions (default: $-1$, thus ignored)

- consider_maxvert_lambda: $\Lambda$ starting from which the stopping condition is active (default: $-1.0$, thus ignored).

After performing an Euler step from $\Lambda$ to $\Lambda + \mathrm{d}\Lambda$, the next step-width $\mathrm{d}\Lambda$ is calculated as $\mathrm{d}\Lambda = \max[\min(B_{\mathrm{fac}}, B_{\mathrm{fac-sc}}), b]$, i.e., taking into account all the upper and lower bounds defined above. Moreover, the multiple halting conditions are checked and the return value of diverge_euler_next is chosen accordingly.

## B.2 Encoding of real harmonics: real_harmonics_t

The enumeration real_harmonics_t encodes each real harmonic up to the $g$-shell such that the orbitals can be easily set up for use as function in site_descr_t. Following Refs. [75, 76], we define real harmonics as

$$
S_{l,m} = \begin{cases} \frac{(-1)^m}{\sqrt{2}}(Y_{l,m} + (-1)^m Y_{l,-m}) & m > 0, \\ Y_{l,0} & m = 0, \\ \frac{1}{\sqrt{2}i}(Y_{l,m} - (-1)^m Y_{l,-m}) & m < 0 \end{cases}
\tag{19}
$$

in terms of the spherical harmonics $Y_{l,m}$. Our naming convention is orb_ followed by the letter of the shell (s,p,d,f,g) and the value of m where the negative values are indicated by an 'm' and positive values by an underscore '_'. Alternatively, all harmonics up to the $d$-shell are accessible by their name.

```
typedef enum real_harmonics_t {
    orb_s = 0,

    orb_pm1 = 1,
    orb_p_0 = 2,
    orb_p_1 = 3,
    orb_py = 1,
    orb_pz = 2,
    orb_px = 3,

    orb_dm2 = 4,
    orb_dm1 = 5,
    orb_d0 = 6,
    orb_d1 = 7,
    orb_d2 = 8,
```

```
16    orb_dxy = 4,
17    orb_dyz = 5,
18    orb_dz2 = 6,
19    orb_dxz = 7,
20    orb_dx2y2 = 8,
21
22    orb_fm3 = 9,
23    orb_fm2 = 10,
24    orb_fm1 = 11,
25    orb_f_0 = 12,
26    orb_f_1 = 13,
27    orb_f_2 = 14,
28    orb_f_3 = 15,
29
30    orb_gm4 = 16,
31    orb_gm3 = 17,
32    orb_gm2 = 18,
33    orb_gm1 = 19,
34    orb_g_0 = 20,
35    orb_g_1 = 21,
36    orb_g_2 = 22,
37    orb_g_3 = 23,
38    orb_g_4 = 24
39 } real_harmonics_t;
```

## C   File Formats

divERGe uses binary files as output for the model and postprocessing data[21]. Their file formats are defined below. Python classes that read divERGe output files are distributed in the `diverge.output` library. Note that we are well aware of the HDF5 library [77], but for high portability and reduction of dependencies decided against using it and instead implemented a very lightweight I/O based on the C standard library functions `fopen()`, `fwrite()`, `fclose()` (or their MPI counterparts).

### C.1   Model output file

To control the contents of the model file beyond their defaults, we provide the structure `diverge_model_output_conf_t`, which is defined as

```
1 typedef struct diverge_model_output_conf_t {
2     int kc;
3     int kf;
4     int kc_ibz_path;
5     int kf_ibz_path;
6     int H;
7     int U;
8     int E;
9     int npath;
10 } diverge_model_output_conf_t;
```

The individual elements control optional storage of (and are treated as boolean variables internally if not specified else)

- `kc`: store coarse momentum mesh

- `kf`: store fine momentum mesh

- `kc_ibz_path`: if an IBZ-path has been set, stores the points in the coarse mesh on the path

---

[21]No output is generated for flow information, as the user themselves are responsible for looping over integration steps, i.e., flowing.

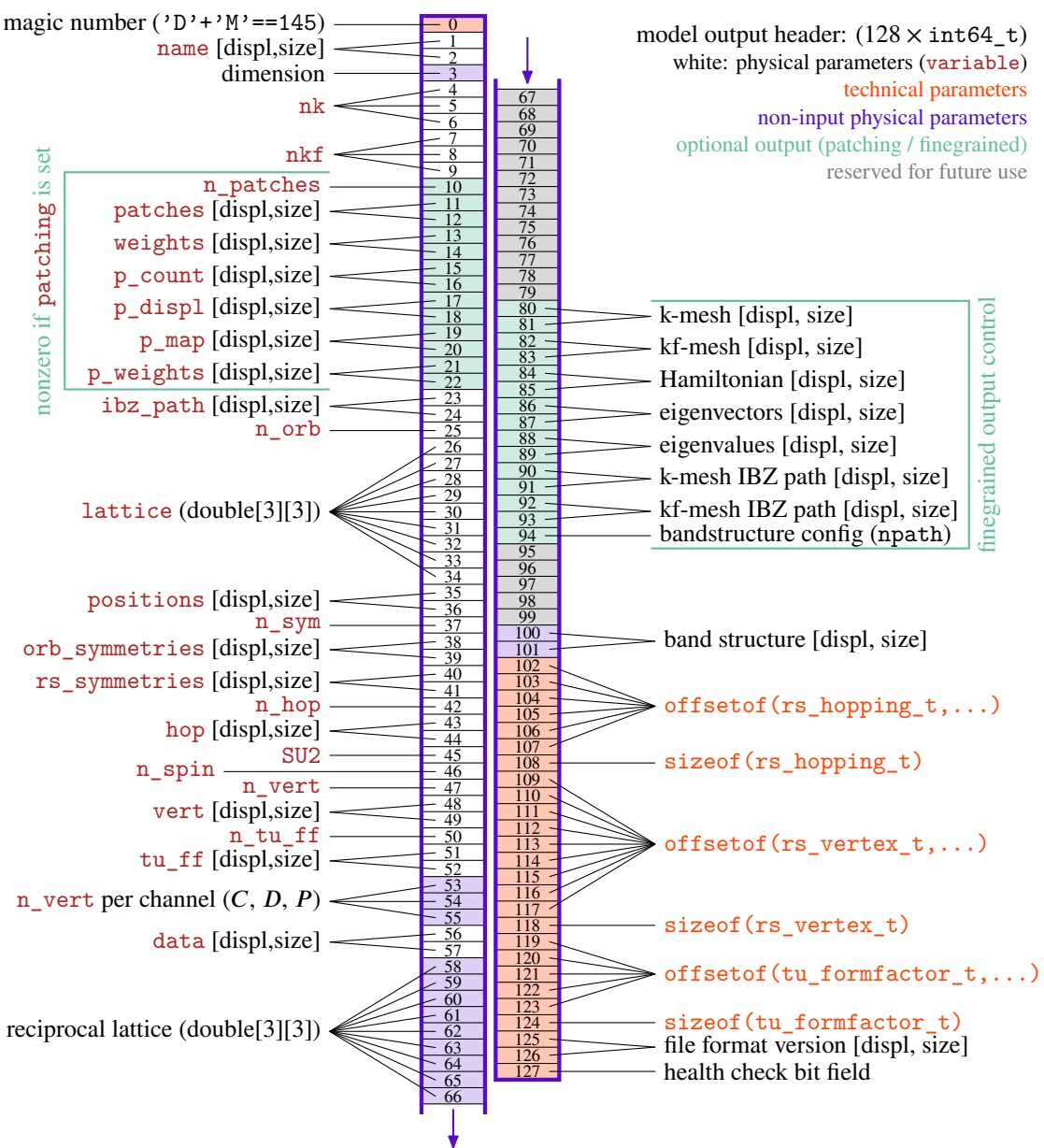

Figure 3: File format header for the files written by `diverge_model_output`. The data is preceded by a header (written in binary form) consisting of 128 64-bit (signed) integer numbers (`index_t`). If not specified else, all header variables are of the `index_t` type. The data section following the header contains all arrays specified through their displacement (displ) and size (given in bytes). Datatypes vary according to the array: `char` (`name`, `data`, file format version), `double` (`weights`, `p_weights`, `ibz_path`, `positions`, `rs_symmetries`, k-mesh, kf-mesh, eigenvalues, band structure), `index_t` (`patches`, `p_count`, `p_displ`, `p_map`), `rs_vertex_t`, `rs_hopping_t`, `tu_formfactor_t`, and `complex128_t` (everything else). More specific information on the array shapes is found in the `diverge.output` library.

- `kf_ibz_path`: if an IBZ-path has been set, stores the points in the fine mesh on the path

- `H`: store the Hamiltonian on the fine mesh

- `U`: store the orbital-to-band transformations on the fine mesh

- `E`: store energies on the fine mesh

- `npath`: **integer value**. If nonzero, use this value as if `diverge_output_set_npath` had been called, but with precedence. Allows control over the number of points on the band structure path (if $> 0$), or, using the internal fine k-mesh and the path constructed from there (cf. `kf_ibz_path`, if $-1$).

The binary model output file consists of a $128 \times 64$ bit header followed by data. A graphical representation of the file format for model output (i.e. its header) is documented in Fig. 3. To read an array from the output file, one must look up the displacement and size from the header. A simple C example program reading an array from a divERGe model file is shown in Listing 3.

```c
#include <stdio.h>
#include <stdint.h>
#include <stdlib.h>

int main(void) {
    FILE* f = fopen("model.dvg", "rb");

    // header
    int64_t header[128];
    fread(header, sizeof(int64_t), 128, f);

    // we want to read the array specified by [88, 89]
    double* energies = (double*)malloc(header[89]); // allocate memory (size)
    fseek(f, header[88], SEEK_SET); // set offset
    fread(energies, 1, header[89], f); // read

    // do something with the array here
    printf("read energiy array of size (%lu)\n", header[89]/sizeof(double));

    free(energies);
    fclose(f);
}
```

Listing 3: C example program that reads the dispersion array from a divERGe model output file (`model.dvg`) to illustrate how to deal with the file format from languages that are not Python.

## C.2 Postprocessing output files

The heterogenous nature of postprocessing options for the different backends suggests specific file formats for each of them. A definition of the respective file headers is given in the following subsections. To control what will be stored, we provide the `diverge_postprocess_conf_t` struct, which is defined as

```c
typedef struct {
    bool    patch_q_matrices;
    bool    patch_q_matrices_use_dV;
    int     patch_q_matrices_nv;
    double  patch_q_matrices_max_rel;
    char    patch_q_matrices_eigen_which;
```

```
7     bool    patch_V;
8     bool    patch_dV;
9     bool    patch_Lp;
10    bool    patch_Lm;
11
12    char    grid_lingap_vertex_file_P[MAX_NAME_LENGTH];
13    char    grid_lingap_vertex_file_C[MAX_NAME_LENGTH];
14    char    grid_lingap_vertex_file_D[MAX_NAME_LENGTH];
15    int     grid_n_singular_values;
16    bool    grid_use_loop;
17    char    grid_vertex_file[MAX_NAME_LENGTH];
18    char    grid_vertex_chan;
19
20    char    tu_which_solver_mode;
21    double  tu_storing_threshold;
22    bool    tu_storing_relative;
23    index_t tu_n_singular_values;
24    bool    tu_lingap;
25    bool    tu_susceptibilities_full;
26    bool    tu_susceptibilities_ff;
27    bool    tu_selfenergy;
28    bool    tu_channels;
29    bool    tu_symmetry_maps;
30 } diverge_postprocess_conf_t;
```

- `patch_q_matrices`: assemble a list of all possible momentum transfers $q_X$ in each of the interaction channels $X$ and save the vertices $V(q_X, k_X, k_{X'})$ in this representation (default: `true`)

- `patch_q_matrices_use_dV`: use the differential vertex instead of the vertex for generation of the `q_matrices` (default: `false`)

- `patch_q_matrices_nv`: how many eigenvectors to store in the `q_matrices`. If <0, do not perform an eigen decomposition, if ==0, store all eigenvectors. (default: 0)

- `patch_q_matrices_max_rel`: restrict the analysis only to those $q_X$ where the relative vertex norm is greater than this value (default: 0.9)

- `patch_q_matrices_eigen_which`: choose which eigenvalues and -vectors to store: 'M'agnitude, 'P'ositive, 'N'egative, or 'A'lternating (default: 'M')

- `patch_V`: include the full vertex in the output (default: `false`)

- `patch_dV`: include the differential vertex in the output (default: `false`)

- `patch_Lp`: include the particle-particle loop in the output (default: `false`)

- `patch_Lm`: include the particle-hole loop in the output (default: `false`)

- `grid_lingap_vertex_file_P`: store the $q_P = 0$ vertex to this file if not "" (default: "")

- `grid_lingap_vertex_file_C`: store the $q_C = 0$ vertex to this file if not "" (default: "")

- `grid_lingap_vertex_file_D`: store the $q_D = 0$ vertex to this file if not "" (default: "")

- `grid_n_singular_values`: Number of singular values to be stored from the lineaized gap solution (default: 20)

- `grid_use_loop`: Solve linearized gap equation for vertex times loop (default: `true`)

- `grid_vertex_file`: name of the file into which the vertex should be stored. ***Requires a lot of disk space!*** (default: `""` — which means nothing will be saved)

- `grid_vertex_chan`: if `grid_vertex_file` is non-empty chooses the channel in which the vertex is stored (options are `'P'`,`'C'`,`'D'`,`'V'`, default: `'0'`)

- `tu_which_solver_mode`: specifies routine to diagonalize channel — possible values are `'e'` (force eigensolver), `'s'` (force SVD) and `'a'` (auto). The default is `'a'` which checks whether the channel at $q$ is hermitian and then decides whether to use an eigenvalue decomposition or an SVD

- `tu_storing_threshold`: absolute value above which eigenvalues/singular values are stored (default: 50, should be smaller or equal to `maxvert`)

- `tu_storing_relative`: consider `tu_storing_threshold` as a relative maximum instead of an absolute value, i.e., store all eigen/singular values and vectors if they are above the product of `tu_storing_threshold` and the maximum eigen/singular value for the channel over all $q$ (default: `false`)

- `tu_n_singular_values`: number of singular values stored from the solution of the linearized gap equation (default: 1)

- `tu_lingap`: evaluate linearized gap equation in each channel (default: `true`)

- `tu_susceptibilities_full`: calculate susceptibility in orbital basis $q, o_{1-4}, s_{1-4}$ (default: `false`)

- `tu_susceptibilities_ff`: calculate susceptibility in TU-native mixed orbital/bond basis $q, o_{1,3}, b_{1,3}, s_{1-4}$ (default: `false`)

- `tu_selfenergy`: store selfenergy if included in the calculation (default: `false`)

- `tu_channels`: store full channels from tu simulation ***Requires a lot of disk space!*** (default `false`)

- `tu_symmetry_maps`: store symmetry transformation for one vertex leg (i.e., eigenvector; default `false`)

### C.2.1 Grid FRG backend

The (binary) grid output file consists of a $64 \times 64$ bit header that is described in Fig. 4. This header is followed by the output data.

### C.2.2 $N$-patch backend

The (binary) $N$-patch output file consists of a $128 \times 64$ bit header that is described in Fig. 5. This header is followed by the output data.

### C.2.3 TUFRG backend

The (binary) TUFRG output file consists of a $128 \times 64$ bit header that is described in Fig. 6. This header is followed by the output data.

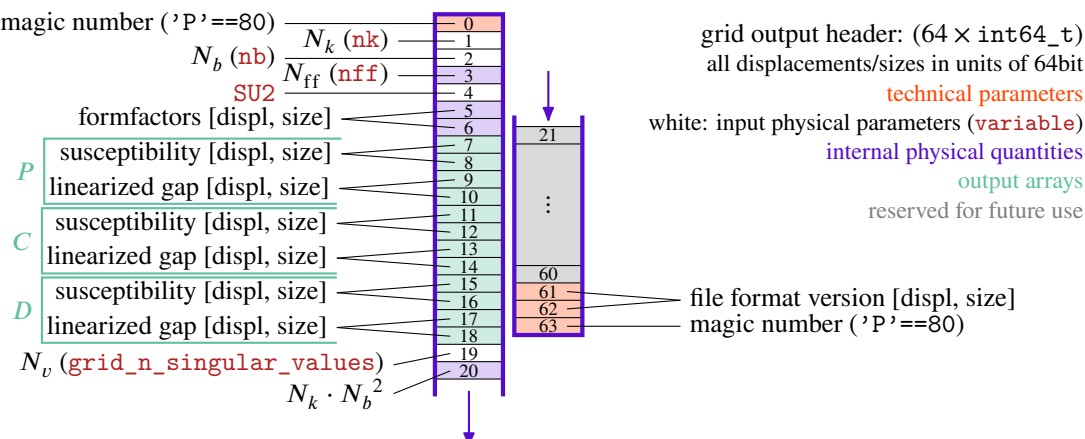

Figure 4: Specification of the file format for grid-FRG postprocessing output files. The header (64 signed 64-bit integers) is followed by data indexed by the displacement/size information. Note that, unlike model output files (cf. Fig. 3), displacement and size information is given in units of 64 bits, i.e., 8 bytes. For grid-FRG, the user has control over saving the susceptibilities for each interaction channel as well as the solutions of a linearized gap euqation at $q_X = 0$.

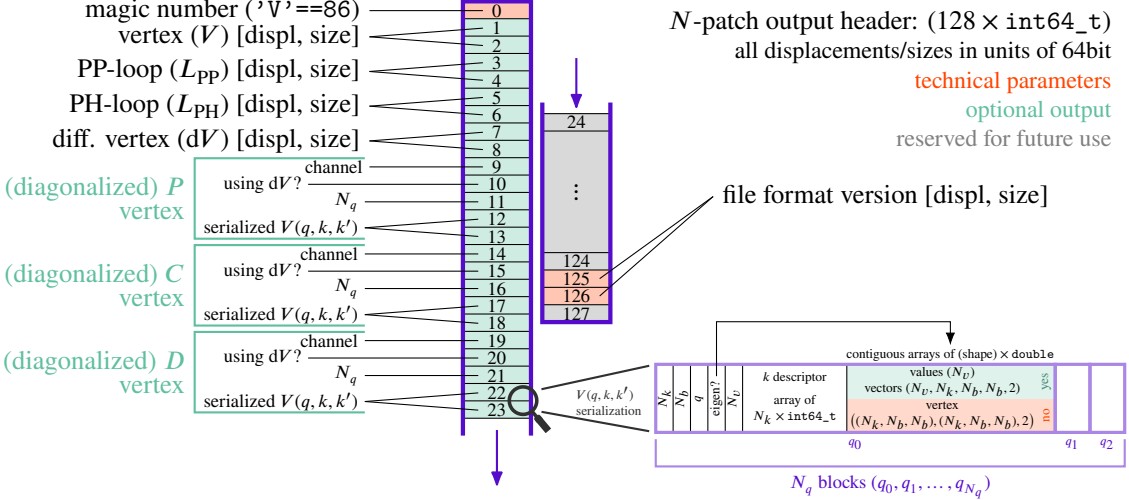

Figure 5: Specification of the file format for $N$-patch FRG postprocessing output files. The header (128 signed 64-bit integers) is followed by data indexed by the displacement/size information. Note that, unlike model output files (cf. Fig. 3), displacement and size information is given in units of 64 bits, i.e., 8 bytes. For $N$-patch FRG, the user has control over saving the vertex ($V$), the loops ($L_{PP}$, $L_{PH}$), the differential vertex ($dV$), and a channel native representation of the vertices optionally eigen-decomposed. As the $N$-patch representation of channel native interactions is non-trivial, the data is further serialized with a sketch on how to read the individual matrices or eigenvalues/-vectors on the bottom right. The Python library `diverge.output` automatically deserializes those objects.

# D  Implementational details

This chapter specifies design choices for the critical parts of each of the backends. We do not discuss the implementation itself, but rather the basic ideas. Furthermore, the parallelization

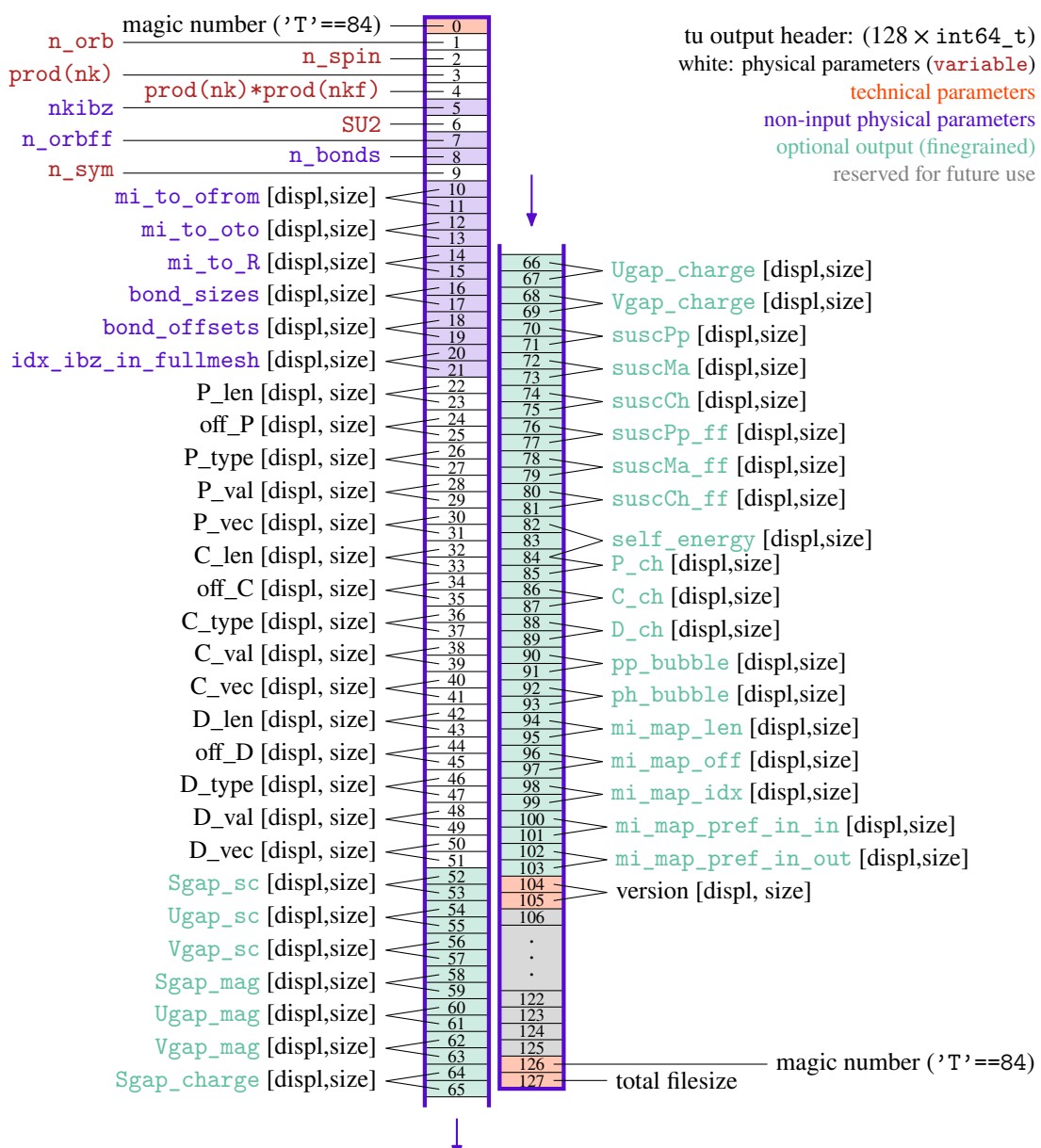

Figure 6: File format header for the file written by `diverge_postprocess_and_write` if TU backend is used. The data is preceded by a header (written in binary form) consisting of 128 64-bit (signed) integer numbers (`index_t`). If not specified else, all header variables are of the `index_t` type. The data section following the header contains all arrays specified through their displacement (displ) (given in bytes) and size (given in number of elements with the respective type). All arrays are of `complex128_t` type but `mi_to_ofrom`, `mi_to_oto`, `mi_to_R`, `bond_sizes`, `bond_offsets`, `idx_ibz_in_fullmesh`, all arrays containing `off` or `len` in their name and `mi_map_idx` which are of type `index_t`. More specific information on the array shapes is found in the `diverge.output` library.

strategy is explained for each of the backends.

## D.1   Grid FRG backend

Conceptually, the process of solving the FRG flow equations on a fixed momentum Bravais grid for all vertex indices is simple. Achieving sufficient performance for simulations on nontrivial two- and three-band models requires splitting the vertex buffers on more than one computation node (via MPI). The main bottleneck is then given by the available memory size, but also by computation of the contractions. For parallelization, the vertices are reordered such that a channel-native representation is obtained and the contractions reduce to many matrix products (along $q$) that are distributed across all compute nodes with MPI, then branching to cuBLAS (GPU) or parallel BLAS (CPU) calls. The reordering procedure, while simple on one node, involves multiple communication steps on a multi-node architecture.

Fundamentally the Grid backend scales with $\mathcal{O}(N_k^4 N_o^6 N_s^6)$ in computation time and with $\mathcal{O}(N_k^3 N_o^4 N_s^4)$ in memory, with bottlenecks usually being communication, memory size, and batched matrix matrix products.

## D.2   $N$-patch backend

$N$-patch FRG inherently breaks momentum conservation and thereby rotational symmetries of the system, which is due to the structure of the method itself: If all momenta are fixed to the Fermi surface $k_{1,2,3}$, the fourth momentum is not always. One approximates the vertex element with one momentum away from the Fermi surface by fixing it to the closest point on the Fermi surface.

The patching procedure and pinning of momentum points to a grid is discussed in Appendix A.1.

Fundamentally the $N$-patch backend scales with $\mathcal{O}(N_p^4 N_o^6 N_s^6)$ in computation time and with $\mathcal{O}(N_p^3 N_o^4 N_s^4)$ in memory, with $N_p$ the number of momentum patches. Since the objects involved are not extremely huge in memory, they are copied onto all MPI ranks of a calculation. The computational work is distributed using MPI and OpenMP on CPUs as well as CUDA on GPUs, with the main issue on both architectures being memory locality (since the vertex products cannot be written as matrix multiplications).

## D.3   TUFRG backend

For the TUFRG backend, the most expensive components for usual models are the loop calculation and the projections. For the projections, the implementation follows Ref. [23], with a slight improvement — the calculation of the expression is split into two parts. We recapitulate the general formula for the projection from $C$ to $P$:

$$\hat{P}[\hat{C}^{-1}[C]]_{o_1,o_3}^{b_1,b_3}(q_P)_{s_1,s_3}^{s_2,s_4} = \delta_{o_1+b_1,o_3+b_3'}\,\delta_{o_3+b_3,o_1+b_1'}\,\delta_{B_1-B_1',B_3'-B_3}\,e^{i(B_3'-B_3)q_P}\,C_{o_1,o_3}^{b_1',b_3'}(B_3'-B_3)_{s_1,s_3}^{s_4,s_2}\,. \tag{20}$$

Here, we already inserted the form-factor bonds. In a first step, we calculate $C_{o_1,o_3}^{b_1',b_3'}(B_3'-B_3)_{s_1,s_3}^{s_4,s_2}$, which technically is a Fourier Transformation onto a restricted mesh. In a second step we multiply out the the vertex in real-space with the prefactors. These are calculated in advance and require only little memory. In the standard form this is parallelized with OpenMP; when enabling MPI, the calculation is split along different $q$ and at the end of step one a call to `MPI_Allreduce` is performed. If GPUs are available, the second step is performed with a hand-written kernel.

For the calculation of the loop we utilize a Fourier transformation trick [55, 78, 79] in

combination with a sharp frequency cutoff. This leads to

$$
L^{ph;b_1,b_3}_{o_1,o_3}(\boldsymbol{q}_X)^{s_4,s_2}_{s_1,s_3} = \sum_{\boldsymbol{R}} e^{-i\boldsymbol{R}\boldsymbol{q}_X} \Big[ G^{s_1,s_3}_{o_1;o_3}(\Lambda, -\boldsymbol{R} - \boldsymbol{B}_1 + \boldsymbol{B}_3) G^{s_2,s_4}_{o_1+b_1;o_3+b_3}(\Lambda, \boldsymbol{R})
$$
$$
+ G^{s_1,s_3}_{o_1;o_3}(-\Lambda, -\boldsymbol{R} - \boldsymbol{B}_1 + \boldsymbol{B}_3) G^{s_2,s_4}_{o_1+b_1;o_3+b_3}(-\Lambda, \boldsymbol{R}) \Big]. \quad (21)
$$

and for the particle-particle channel

$$
L^{pp;b_1,b_3}_{o_1,o_3}(\boldsymbol{q}_P)^{s_2,s_4}_{s_1,s_3} = \sum_{\boldsymbol{R}} e^{-i\boldsymbol{R}\boldsymbol{q}_P} \Big[ G^{s_1,s_3}_{o_1;o_3}(-\Lambda, \boldsymbol{R} - \boldsymbol{B}_1 + \boldsymbol{B}_3) G^{s_2,s_4}_{o_1+b_1;o_3+b_3}(\Lambda, \boldsymbol{R})
$$
$$
+ G^{s_1,s_3}_{o_1;o_3}(\Lambda, \boldsymbol{R} - \boldsymbol{B}_1 + \boldsymbol{B}_3) G^{s_2,s_4}_{o_1+b_1;o_3+b_3}(-\Lambda, \boldsymbol{R}) \Big], \quad (22)
$$

with the usual definition of the Green's function as

$$
G^{-1} = (G_0^{-1} - \Sigma)^{-1}. \quad (23)
$$

Again, the implementation uses OpenMP for parallelization over orbitals, spins and bonds. An optimized version for MPI (based on FFTW-MPI) is provided at compile time. In case of GPU usage, a hand written kernel is used for the calculation of the Green's function products and combined with calls to cuFFT.

The flow product is evaluated using GEMM calls, which are distributed along the coarse momentum index and offloaded to GPUs if available. Depending on the chosen model and parameters, the leading scaling is either the loop $\mathcal{O}(N_q N_k \log(N_q N_k) N_{ff}^2 N_o^2 N_s^4)$, the flow step matrix products $\mathcal{O}(N_q N_{ff}^3 N_o^3 N_s^6)$ or the the interchannel projections $\mathcal{O}(N_o^2 N_{ff}^3 N_s^4 N_q)$. Furthermore, memory usage of a calculation scales with $\mathcal{O}(N_q N_o^2 N_{ff}^2 N_s^4)$.

Analogously, we use Fourier transformations for the calculation of the static self-energy,

$$
\frac{d\Sigma^{s_1;s_3}_{o_1;o_3}(\boldsymbol{R},0)}{d\Lambda} = -\frac{1}{2\pi} \sum_{\nu=\pm\Lambda} \Big[ G^{s_2;s_4}_{o_2;o_4}(\boldsymbol{R} + \boldsymbol{B}_1 - \boldsymbol{B}_3, \nu) \delta_{o_1+b_1,o_2} \delta_{o_3+b_3,o_4} P^{b_1,b_3}_{o_1,o_3}(\boldsymbol{R} + \boldsymbol{B}_1 - \boldsymbol{B}_3, \nu)^{s_2;s_4}_{s_1;s_3}
$$
$$
+ G^{s_2;s_4}_{o_2;o_4}(-\boldsymbol{R} - \boldsymbol{B}_1 + \boldsymbol{B}_3, \nu) \delta_{o_1+b_1,o_4} \delta_{o_3+b_3,o_2} C^{b_1,b_3}_{o_1,o_3}(-\boldsymbol{R} - \boldsymbol{B}_1 + \boldsymbol{B}_3, -\nu)^{s_4;s_2}_{s_1;s_3}
$$
$$
+ \delta_{o_1+b_1,o_3} \delta_{o_4+b_4,o_2} \delta_{\boldsymbol{B}_1,-\boldsymbol{R}} e^{i\boldsymbol{k}'\boldsymbol{B}_4} G^{s_2;s_4}_{o_2;o_4}(\boldsymbol{k}', \nu) D^{b_1,b_4}_{o_1,o_4}(0)^{s_3;s_2}_{s_1;s_4} \Big]. \quad (24)
$$

### D.4   Scaling and runtime

To give potential users a better feeling of what can and cannot be done with divERGe we present scaling plots and give reference runtimes for a three band Emery model. The form-factor cutoff distance is set to 1.2 and the number of coarse and fine momentum points is varied, see Fig. 7. We observe a significant speedup when computing on GPUs, though the GPU speedup is not optimal (bottlenecks: host-device communication and parts of the algorithms that run on CPUs). The CPU algorithm has its main bottleneck in communication, precisely in the calculation of the loop (via FFTW-MPI).

Note that the momentum resolutions treated in Fig. 7 are way beyond what is necessary for most calculations. A fully converged (3 orbitals, 127 formfactors, $32^2$ momentum points and $(32 \times 55)^2$ integration points for the loop) standard calculation (reaching $\Lambda = 10^{-3}$) on a single JURECA DC-GPU node (4 NVIDIA A100 GPUs) takes around 5 Minutes.

## E   Example codes used in this manuscript

The data used in Fig. 1 (and Listing 1) were generated using the Python code given in Appendix E.1. Figure 2 (plotted via Listing 2) contains data generated using the C++ code from

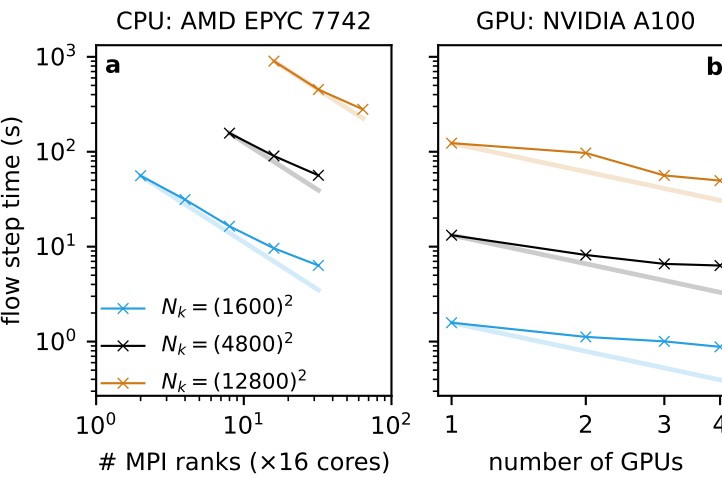

Figure 7: Performance analysis of the TU$^2$FRG backend of divERGe for a three-orbital model. We vary the total number of momentum points ($N_k$) and the number of MPI ranks (a) or GPUs (b). The computational cost for large $N_k$ is dominated by FFTs, i.e. $\mathcal{O}(N_k \log N_k) \approx \mathcal{O}(N_k)$ (faint lines). The largest system (orange) exceeds on-GPU memory and hence suffers slightly from host-device data transfer.

Appendix E.2. Both codes are part of the online examples in divERGe's git repository [18]. The code used in the example blocks is available in the git repository as `examples/c_cpp/honeycomb_tutorial.cpp`. The code for the performance analysis in Fig. 7 can be found in `performance/t2g_model.c`.

## E.1  Emery model (Python)

```python
#!/usr/bin/env python3
#coding:utf8
import diverge
import numpy as np

# initialization, needed for MPI compiled library
diverge.init( None, None )

# some parameters
params_emery = dict( Ud=3.0, Up=1.0, tdd=0.2, tpp=0.3, tdp=1.0 )
params_frg = dict( nk=8, nkf=5, model_output="model.dvg" )

# we now use the actual model structure
model = diverge.model_init()
model.contents.name = b'Emery'
model.contents.n_orb = 3
model.contents.SU2 = True
model.contents.n_spin = 1
model.contents.nk[0] = params_frg['nk']
model.contents.nk[1] = params_frg['nk']
model.contents.nkf[0] = params_frg['nkf']
model.contents.nkf[1] = params_frg['nkf']

# lattice and positions
lattice = diverge.view_array( model.contents.lattice, dtype=np.float64, shape=(3,3) )
lattice[:,:] = np.eye(3)
positions = diverge.view_array( model.contents.positions, dtype=np.float64, shape=(3,3) )
positions[:,:] = np.array( [[0,0,0], [0.5,0,0], [0,0.5,0]] )

# get the hopping phase between p orbitals
def get_phase( o1, o2, po1, po2 ):
    if o1 == o2 and o1 == 0:
        return 1.0
    elif o1 == 0 and o2 == 1:
        return -1.0 if po2[0] > 0 else 1.0
    elif o1 == 0 and o2 == 2:
```

```
37            return 1.0 if po2[1] > 0 else -1.0
38        elif o1 == 1 and o2 == 0:
39            dist = po1-po2
40            return -1.0 if dist[0] > 0 else 1.0
41        elif o1 == 2 and o2 == 0:
42            dist = po1-po2
43            return 1.0 if dist[1] > 0 else -1.0
44        else:
45            dist = po2-po1
46            if dist[0] > 0:
47                return -1.0 if dist[1] > 0 else 1.0
48            else:
49                return -1.0 if dist[1] < 0 else 1.0
50
51 # hopping parameters
52 hop_list = []
53 tdd_norm = 1.0
54 tdp_norm = 0.5
55 tpp_norm = 0.5*np.sqrt(2.)
56 for Rx in range(-2,3):
57     for Ry in range(-2,3):
58         for o1 in range(3):
59             for o2 in range(3):
60                 dist = np.linalg.norm(lattice[0]*Rx + lattice[1]*Ry +
    positions[o2] - positions[o1])
61                 phase = get_phase(o1,o2,positions[o1],lattice[0]*Rx + lattice
    [1]*Ry + positions[o2])
62                 if (np.isclose(dist, tdd_norm) and o1 == 0 and o2 == 0):
63                     hop_list.append( ( [Rx, Ry, 0], o1, o2, 0,0, (-
    params_emery['tdd']*phase,0.0) ) )
64                 if (np.isclose(dist, tpp_norm) and o1 != 0 and o2 != 0):
65                     hop_list.append( ( [Rx, Ry, 0], o1, o2, 0,0, (-
    params_emery['tpp']*phase,0.0) ) )
66                 if (np.isclose(dist, tdp_norm)):
67                     hop_list.append( ( [Rx, Ry, 0], o1, o2, 0,0, (-
    params_emery['tdp']*phase,0.0) ) )
68 hop_array = diverge.alloc_array( (len(hop_list),), dtype="rs_hopping_t" )
69 hop_array[:] = hop_list
70 diverge.mpi_py_eprint( "using %i hopping elements" % len(hop_list) )
71 # set the hopping pointer
72 model.contents.hop = hop_array.ctypes.data
73 model.contents.n_hop = hop_array.size
74
75 # vertices
76 vertices = diverge.alloc_array( (3,), dtype="rs_vertex_t" )
77 vertices[0] = ( 'D', [0,0,0], 0,0, -1,0,0,0, (params_emery['Ud'],0.0) )
78 vertices[1] = ( 'D', [0,0,0], 1,1, -1,0,0,0, (params_emery['Up'],0.0) )
79 vertices[2] = ( 'D', [0,0,0], 2,2, -1,0,0,0, (params_emery['Up'],0.0) )
80 # set the vertex pointer
81 model.contents.vert = vertices.ctypes.data
82 model.contents.n_vert = vertices.size
83
84 # set the irreducible path
85 ibz_path = diverge.view_array( model.contents.ibz_path, dtype=np.float64,
    shape=(4,3) )
86 model.contents.n_ibz_path = 4
87 ibz_path[:,:] = np.array( [ [0,0,0], [0,0.5,0], [0.5,0.5,0], [0,0,0] ] )
88
89 # validate the model
90 if diverge.model_validate( model ):
91     diverge.mpi_py_eprint( "invalid model!" )
92     diverge.mpi_exit(1)
93
94 # and initialize internal structures
95 diverge.model_internals_common( model )
96
97 # write to file
98 checksum = diverge.model_to_file_PY( model, params_frg['model_output'],
    kf_ibz_path=1, npath=-1 )
99 diverge.mpi_py_eprint( "wrote model to file %s (%s)" % (params_frg['
    model_output'], checksum) )
100
```

```
101 # free resources
102 diverge.model_free( model )
103
104 # we need to accept that MPI requires finalization
105 diverge.finalize()
```

## E.2   Honeycomb model (C++)

```cpp
1 #include <diverge.h>
2 #include <diverge_Eigen3.hpp>
3
4 const index_t nk = 1200;
5 const double t = 1.0;
6 const double tp = 0.1;
7 const double U = 3.6;
8 const double V = 0.0;
9 const double V2 = 0.0;
10 const index_t np = 4;
11
12 int main(int argc, char** argv) {
13     // init
14     diverge_init( &argc, &argv );
15     mpi_loglevel_set( 5 );
16     diverge_compilation_status();
17     diverge_model_t* m = diverge_model_init();
18
19     // set up honeycomb lattice
20     m->lattice[0][0] = m->lattice[1][0] = sin(M_PI/3.);
21     m->lattice[0][1] = m->lattice[1][1] = cos(M_PI/3.);
22     m->lattice[0][1] *= -1.0;
23     m->lattice[2][2] = 1.0;
24     Map<Mat3d> pos(m->positions[0]);
25     Map<Mat3d> lat(m->lattice[0]);
26     pos.col(0) = -1./3. * (lat.col(0) + lat.col(1));
27     pos.col(1) =  1./3. * (lat.col(0) + lat.col(1));
28     m->n_orb = 2;
29     m->n_spin = 1;
30     m->SU2 = 1;
31     m->nk[0] = m->nk[1] = nk;
32     m->nkf[0] = m->nkf[1] = 1;
33     m->n_ibz_path = 4;
34     m->ibz_path[1][0] = 0.5;
35     m->ibz_path[2][0] = 2./3.;
36     m->ibz_path[2][1] = 1./3.;
37     strcpy( m->name, __FILE__ );
38
39     // find hopping parameters
40     double t_norm = pos.col(0).norm();
41     double tp_norm = 1.0;
42     m->hop = (rs_hopping_t*)calloc(sizeof(rs_hopping_t), 1024);
43     m->vert = (rs_vertex_t*)calloc(sizeof(rs_vertex_t), 1024);
44     m->vert[m->n_vert++] = rs_vertex_t{ 'D', {0,0,0}, 0,0, -1,0,0,0, U };
45     m->vert[m->n_vert++] = rs_vertex_t{ 'D', {0,0,0}, 1,1, -1,0,0,0, U };
46     for (int Rx=-5; Rx<=5; ++Rx)
47     for (int Ry=-5; Ry<=5; ++Ry)
48     for (int o=0; o<2; ++o)
49     for (int p=0; p<2; ++p) {
50         double dist = (lat.col(0) * Rx + lat.col(1) * Ry + pos.col(p) - pos.col(o)).norm();
51         if (std::abs(dist - t_norm) < 1e-5) {
52             m->hop[m->n_hop++] = rs_hopping_t{ {Rx,Ry,0}, o,p, 0,0, t };
53             m->vert[m->n_vert++] = rs_vertex_t{ 'D', {Rx,Ry,0}, o,p, -1,0,0,0, V };
54         }
55         if (std::abs(dist - tp_norm) < 1e-5) {
56             m->hop[m->n_hop++] = rs_hopping_t{ {Rx,Ry,0}, o,p, 0,0, tp };
57             m->vert[m->n_vert++] = rs_vertex_t{ 'D', {Rx,Ry,0}, o,p, -1,0,0,0, V2 };
58         }
59     }
60
61     // set up symmetries
62     m->orb_symmetries = (complex128_t*)malloc(sizeof(complex128_t)*12*4);
63     site_descr_t sites[2];
64     for (int s=0; s<2; ++s) {
65         sites[s].amplitude[0] = 1.0;
66         sites[s].function[0] = orb_s;
```

```
67          sites[s].n_functions = 1;
68      }
69      sym_op_t sym;
70      Map<Vec3d> sym_nvec(sym.normal_vector);
71      sym_nvec = Vec3d(0,0,1);
72      sym.type = 'R';
73      for (int i=0; i<6; ++i) {
74          sym.angle = 60*i;
75          diverge_generate_symm_trafo( 1, sites, 2, &sym, 1, m->rs_symmetries[m
    ->n_sym][0],
76                  m->orb_symmetries+m->n_sym*4 );
77          m->n_sym++;
78      }
79      sym.type = 'M';
80      sym_nvec *= 0.0;
81      for (int i=0; i<6; ++i) {
82          sym_nvec.head(2) = Rot2d(M_PI/6.0 * i).toRotationMatrix() * Vec2d(1,0)
    ;
83          diverge_generate_symm_trafo( 1, sites, 2, &sym, 1, m->rs_symmetries[m
    ->n_sym][0],
84                  m->orb_symmetries+m->n_sym*4 );
85          m->n_sym++;
86      }
87
88      // check!
89      if (diverge_model_validate(m))
90          diverge_mpi_exit(EXIT_FAILURE);
91
92      // internals, patching, and saving the model to disk
93      diverge_model_internals_common(m);
94      diverge_model_set_filling( m, NULL, -1, 0.6 );
95      diverge_model_internals_patch( m, np );
96      diverge_model_output_conf_t cfg = diverge_model_output_conf_defaults_CPP()
    ;
97      cfg.kc = true;
98      cfg.npath = -1;
99      cfg.kc_ibz_path = 1;
100     char* md5 = diverge_model_to_file_finegrained(m, __FILE__ ".mod.dvg", &cfg
    );
101     mpi_usr_printf( "md5sum: %s\n", md5 );
102
103     // flow
104     diverge_flow_step_t* s = diverge_flow_step_init( m, "patch", "PCD" );
105     diverge_euler_t eu = diverge_euler_defaults_CPP();
106     eu.Lambda = 10.0;
107     eu.dLambda = -1.0;
108     eu.maxvert = 10.0;
109     eu.dLambda_fac = 1.0;
110     eu.dLambda_fac_scale = 0.1;
111     double maxvert = 0;
112     double chanmax[3] = {0};
113     do {
114         diverge_flow_step_euler( s, eu.Lambda, eu.dLambda );
115         diverge_flow_step_vertmax( s, &maxvert );
116         diverge_flow_step_chanmax( s, chanmax );
117         printf( "%.5e %.5e %.5e %.5e %.5e\n", eu.Lambda, chanmax[0], chanmax
    [1], chanmax[2], maxvert );
118     } while (diverge_euler_next( &eu, maxvert ));
119
120     // post processing
121     diverge_postprocess_and_write( s, __FILE__ ".out.dvg" );
122
123     // cleanup
124     diverge_flow_step_free( s );
125     diverge_model_free( m );
126     diverge_finalize();
127 }
```

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
