# Peer review of "divERGe implements various Exact Renormalization Group examples"

_SciPost Physics Codebases, doi:SciPost Phys. Codebases 26 (2024) , SciPost Phys. Codebases 26-r0.5 (2024)_

## Round 1 · Referee Report · Anonymous · 2023-12-4

Strengths

- the technical presentation and writing of the paper is excellent
- the software package seems very sophisticated and flexible
- documentation of the code appears fairly complete and exhaustive
- package interfaces with established ab-initio codes like Wannier90

Weaknesses

- claim of "high-performance", but no benchmark or scaling plot is shown
- example scripts in the appendix are commented but not well-explained
- paper seems strongly expert-oriented in the current stage
- fRG is advocated as replacement of RPA for real materials, but on what basis?

Report

The paper "divERGe implements various Exact Renormalization Group examples" by Hauck et al., describes the implementation of the divERGe software, a C++/python library for performing different types of momentum space functional renormalization group calculations, with an interface to standard ab-initio codes. Such an implementation is very welcome, as it allows fRG newcomers to set up their own calculations straightforwardly if an appropriate model Hamiltonian can be provided. The paper is clearly written and the "typology" of the code is detailed in-depth, with even more documentation available online. From my perspective, the manuscript definitely deserves publication, though three points need further clarification:

(1) The package is claimed to be "high-performance", but no scaling plots or benchmarks with similar implementations are shown. Furthermore, but this might be my mistake, I could not find an estimate of how much computational resources are needed to run a typical fRG calculation using divERGe. I think this might be useful for people wanting to use the code.

(2) Though the code is well-documented, I find that the paper is lacking a simple step-by-step example walking potential users through the code. Two example scripts with comments are given in the appendix, but they are, at least from my perspective, too convoluted and lengthy to be standalone. I would appreciate if the authors could add one of these examples to the main text, split into smaller segments and dressed with additional explanations.

(3) I am slightly concerned about the following: fRG is put forward as an alternative to RPA for realistic material calculations. In my opinion, the arguments brought up in the paper to support this statement are not well-founded. It seems that in the current stage of the software, the self-energy (apart from approximating it as a constant) is discarded, and bold propagators are thus replaced by bare, unrenormalized ones. Therefore, neither coherent Landau quasiparticles nor their absence seems to be captured and, for example, metal-to-insulator transitions are elusive in the implemented methods. Maybe this is not so important, as the principal goal of the fRG method in this form is the determination of order in the ground state. But even then, I have my doubts that quantitative results for real materials can be produced with the code. Assume, for example, a superconducting ground state. One would expect that it emerges from an effective attractive interaction between sufficiently long-lived quasiparticles, but, as argued above, these are not included in the current methodology. Thus, if quantitative accuracy is not the goal, what is the real advantage of fRG over RPA or FLEX approximations, which seem numerically cheaper to pursue? The authors mention that fRG provides additional screening, but not fully self-consistently and, due to the neglect of frequency dependencies, retardation effects are missing entirely. My question thus is, how much do the fRG predictions actually differ from RPA or FLEX when applied to the DFT bandstructures?

On a side note: If the Fermi surface is not renormalized by self-energy insertions the definition of the filling per site seems sketchy to me. The code is supposed to treat interacting quantum systems so the electronic density is not generally equivalent to that of the non-interacting system. Should this not be computed self-consistently at least in Hartree-Fock approximation if a constant self-energy can be included?

Requested changes

- add scaling plots and resource estimates to support claim of "high-performance"
- add step-by-step example, maybe using one of the scripts from the appendix
- revise the motivation for using the implemented fRG methods for materials

---

## Round 1 · Referee Report · Claudio Attaccalite · 2023-12-18

Strengths

-the paper is clear and guide the user step-by-step in the use of the code
-example are provided

Weaknesses

-discussion on parallelization and scaling of the code can be enlarged

Report

The manuscript presents a spet-by-spet use of the divERGe code for the exact renormalisation group. As it was a bit complicated to find additional reviewers, I decided to proceed with one report, plus some additional questions/remarks I would like to pose to the authors:

1) In the manuscript, parallelisation is barely mentioned in one or two places. Can the authors be more specific about what is being parallelized and how, and what part is GPU accelerated?

2) The authors mention the difficulty of available memory when parallelizing in MPI, have they considered parallelizing in openMP?

3) How does the code scale with system size, orbitals, k-points both in terms of computation time and memory usage?

4) Some of the ab-initio packages mentioned at the beginning of the manuscript also include code to estimate the V_{1,2,34} parameters, for example HP(https://arxiv.org/abs/2203.15684) in QuantumEspresso. Have the authors considered the possibility of linking their code to something similar?

---

## Round 2 · Referee Report · Anonymous (Referee 1) · 2024-1-19

Report

The author's have properly adressed all my comments and concerns and updated the paper accordingly. I think the manuscript is now ready for publication.

---

## Round 2 · Author Response

Report of Referee A

The manuscript presents a step-by-step use of the divERGe code for the exact renormalisation group. As it was a bit complicated to find additional reviewers, I decided to proceed with one report, plus some additional questions/remarks I would like to pose to the authors:

We gratefully acknowledge the handling of this complicated situation by the editor and are happy to receive their report.

  • In the manuscript, parallelization is barely mentioned in one or two places. Can the authors be more specific about what is being parallelized and how, and what part is GPU accelerated?

We thank the referee for raising this important point, which we tried to improve upon by including some details on the parallelization in Appendix D4. There, we also added a scaling plot for a single flow step of the TUFRG backend (which is expected to be the most widely used one). Note that scaling information was requested by referee two.

  • The authors mention the difficulty of available memory when parallelizing in MPI, have they considered parallelizing in openMP?

We apologize for this misunderstanding. The code is – whenever possible – already parallel using OpenMP on the single node/rank level. We added an MPI algorithm based on the MPI version of FFTW3 for the loop calculation of the TU backend rendering all three backends MPI parallel as well. Note that the GPU implementation of some of the backends may not be favorable in conjunction with MPI usage. We clarified the different levels of parallelization in Appendix D

  • How does the code scale with system size, orbitals, k-points both in terms of computation time and memory usage?

We added the main computational cost for each of the backends in Appendix D.

  • Some of the ab-initio packages mentioned at the beginning of the manuscript also include code to estimate the $V_{1,2,3,4}$ parameters, for example HP (https://arxiv.org/abs/2203.15684) in QuantumEspresso. Have the authors considered the possibility of linking their code to something similar?

We thank the editor for drawing our attention to this library, which we were not aware of. We fully agree that an automated interface would be beneficial, and added a corresponding remark in the conclusion.

Report of Referee B

The paper "divERGe implements various Exact Renormalization Group examples" by Hauck et al., describes the implementation of the divERGe software, a C++/python library for performing different types of momentum space functional renormalization group calculations, with an interface to standard ab-initio codes. Such an implementation is very welcome, as it allows fRG newcomers to set up their own calculations straightforwardly if an appropriate model Hamiltonian can be provided. The paper is clearly written and the "typology" of the code is detailed in-depth, with even more documentation available online. From my perspective, the manuscript definitely deserves publication, though three points need further clarification:

We thank the referee for this kind assessment of our work. We reply to the comments in a point-by-point fashion below.

  • The package is claimed to be "high-performance", but no scaling plots or benchmarks with similar implementations are shown. Furthermore, but this might be my mistake, I could not find an estimate of how much computational resources are needed to run a typical fRG calculation using divERGe. I think this might be useful for people wanting to use the code.

We thank the referee for this remark and added details on parallelization in Appendix D, including scaling plots of the most widely used backend (TUFRG) with both the CPU (MPI) and GPU implementation. Even though providing prototypical runtimes is difficult given the scope of models that can be accessed with divERGe, we added some timing information on state-of-the-art hardware in Appendix D. As we are not aware of any other open-source implementation of any of the backends contained in divERGe, we cannot give performance benchmarks comparing with other codes. We note, however, that the implementation fulfills the correctness benchmark provided in Eur. Phys. J. B (2022) 95:65.

  • Though the code is well-documented, I find that the paper is lacking a simple step-by-step example walking potential users through the code. Two example scripts with comments are given in the appendix, but they are, at least from my perspective, too convoluted and lengthy to be standalone. I would appreciate if the authors could add one of these examples to the main text, split into smaller segments and dressed with additional explanations.

We agree with the referee that the given example scripts in the appendix are rather lengthy and slightly convoluted. To cure this issue, we equipped the manuscript with example boxes containing real-world snippets that, when plugged together, yield a working example that is also part of the example collection of the repository. Furthermore, we added small explanations to each of the snippets to make it easier accessible.

  • I am slightly concerned about the following: fRG is put forward as an alternative to RPA for realistic material calculations. In my opinion, the arguments brought up in the paper to support this statement are not well-founded. It seems that in the current stage of the software, the self-energy (apart from approximating it as a constant) is discarded, and bold propagators are thus replaced by bare, unrenormalized ones. Therefore, neither coherent Landau quasiparticles nor their absence seems to be captured and, for example, metal-to-insulator transitions are elusive in the implemented methods. Maybe this is not so important, as the principal goal of the fRG method in this form is the determination of order in the ground state. But even then, I have my doubts that quantitative results for real materials can be produced with the code. Assume, for example, a superconducting ground state. One would expect that it emerges from an effective attractive interaction between sufficiently long-lived quasiparticles, but, as argued above, these are not included in the current methodology. Thus, if quantitative accuracy is not the goal, what is the real advantage of fRG over RPA or FLEX approximations, which seem numerically cheaper to pursue? The authors mention that fRG provides additional screening, but not fully self-consistently and, due to the neglect of frequency dependencies, retardation effects are missing entirely. My question thus is, how much do the fRG predictions actually differ from RPA or FLEX when applied to the DFT bandstructures?

The referee is correct, we do not strive for quantitative results since this is not possible with the FRG flavor at hand. However, while FRG does not incorporate quasiparticle renormalization, it does include all three diagrammatic channels on equal footing – which is neither done by RPA nor by FLEX. Of course, if it is a priori clear what results to expect, using an RPA calculation is the cheaper solution, since we can just pick the channel we expect to be most relevant – however if multiple channels are competing or no such intuition is available performing a diagramatically unbiased calculation is of utmost importance. Concerning the last question: Especially in multi-site/orbital systems, results from RPA and FRG tend to differ qualitatively due the absence of feedback between the diagrammatic channels. Furthermore, when one aims to run RPA simulations, one can simply select a flow in only a single channel, which is equivalent to an RPA calculation in that channel. The current implementation in divERGe even assures that the computational complexity is the same as (or even better than) for pure RPA.

  • On a side note: If the Fermi surface is not renormalized by self-energy insertions the definition of the filling per site seems sketchy to me. The code is supposed to treat interacting quantum systems so the electronic density is not generally equivalent to that of the non-interacting system. Should this not be computed self-consistently at least in Hartree-Fock approximation if a constant self-energy can be included?

In the beginning of the flow, filling and chemical potential are equivalent, since the model is non-interacting and at $T=0$. As the chemical potential may change when including interactions, the standard formulation included in divERGe does not fix the filling once a flow is started. Only when a quasi-canonical description is explicitly requested by the user, we provide the facilities to set the filling to a given value at each flow step. In response to the referee's note, we added a comment that summarizes the above points in the manuscript.

---

## Round 2 · List of Changes

• Example boxes with real-world snippets that give an actual example code when plugged together
  • Comment on the fact that a quasi-canonical description (i.e. forcing the particle number to be fixed even when self-energies are included) is ill-defined in the flavor of FRG at hand
  • Comment on the integration with interaction parameters from first-principle codes
  • Scaling information for each of the backends in Appendix D.1-D.3
  • Actual scaling plot for the TUFRG backend in both its GPU and CPU (MPI+OpenMP) implementation in Appendix D.4
  • Note on the expected runtime of real-world systems on state-of-the-art hardware

---

## Editorial Decision

published